# Supervised Contrastive Learning from Weakly-Labeled Audio Segments for Musical Version Matching

**Joan Serrà** [1]  **R. Oguz Araz** [2]  **Dmitry Bogdanov** [2]  **Yuki Mitsufuji** [1,3]

## Abstract

Detecting musical versions (different renditions of the same piece) is a challenging task with important applications. Because of the ground truth nature, existing approaches match musical versions at the track level (e.g., whole song). However, most applications require to match them at the segment level (e.g., 20 s chunks). In addition, existing approaches resort to classification and triplet losses, disregarding more recent losses that could bring meaningful improvements. In this paper, we propose a method to learn from weakly annotated segments, together with a contrastive loss variant that outperforms well-studied alternatives. The former is based on pairwise segment distance reductions, while the latter modifies an existing loss following decoupling, hyper-parameter, and geometric considerations. With these two elements, we do not only achieve state-of-the-art results in the standard track-level evaluation, but we also obtain a breakthrough performance in a segment-level evaluation. We believe that, due to the generality of the challenges addressed here, the proposed methods may find utility in domains beyond audio or musical version matching.

## 1. Introduction

When two audio tracks contain different renditions of the same musical piece, they are considered musical versions[1]. Musical versions are inherent in human culture and predate recorded music and notation, as ancient music was transmitted solely through playing and listening (Ball, 2010), which naturally led to variations in tunes, rhythms, structures, etc. Learning representations of musical versions is a challenging task due to the degree and amount of variations that

can be present between versions, which go beyond typical augmentations used by the machine learning community. Two musical versions may feature different instrumentation or timbre, together with tonality and chord modifications, altered melodies, substantial changes to rhythm and tempo, an alternate temporal development or structure, and many more (Yesiler et al., 2021). Yet, musical versions retain their essence, to the point that we can generally agree whether two of them correspond to the same piece or not[2]. Therefore, learnt version representations need to encapsulate multiple characteristics shared between versions that, at the same time, can discriminate them from other pieces.

Musical version matching has several relevant applications (Serrà, 2011; Yesiler et al., 2021), including specific applications to plagiarism and near-duplicate detection[3]. Beyond business impact (Page, 2023) and cultural/artistic appreciation, some applications have become even more relevant today, given the sustained rise and improvement of music generative models (e.g., Copet et al., 2023; Evans et al., 2024; Liu et al., 2024). Indeed, the recent efforts on assessing music data replication, memorization, and attribution in such models exploit some form of music similarity (Barnett et al., 2024; Bralios et al., 2024) or, for improved results, musical version matching (Batlle-Roca et al., 2024).

A fundamental limitation of version matching approaches is that they operate at the full-track level, learning and extracting individual representations from relatively long recordings (for instance, a few-minute song). This is due to ground truth version annotations being only available per track. However, the segments of interest, for both classical and modern applications, are much shorter than the track length (for instance, around 10–20 s). This mismatch between the learning and inference stages, as we will see, causes a dramatic performance degradation (Sec. 5). Another challenge is that, in contrast to standard supervised learning tasks, musical version data sets contain only a few items per class.

---

[1]Sony AI [2]Music Technology Group, Universitat Pompeu Fabra [3]Sony Group Corporation. Correspondence to: Joan Serrà <joan.serra@sony.com>.

*Proceedings of the 42nd International Conference on Machine Learning*, Vancouver, Canada. PMLR 267, 2025. Copyright 2025 by the author(s).

[1]A related but more restrictive and biased term in the literature is "cover songs". To better understand this restriction and bias, see the discussion found in, for example, Yesiler et al. (2021).

[2]A well-known source collecting this information is https://secondhandsongs.com.

[3]Note that musical version matching may expand and subsume traditional music fingerprinting (Cano et al., 2005).

*Table 1.* Comparison of characteristics for a number of existing approaches and the proposed method CLEWS. We exclude multi-feature and/or multi-modal approaches (for example fusing CQT and melody estimations or leveraging audio and lyrics information). For further details and approaches we refer to the survey by Yesiler et al. (2021).

| NAME(S) | MAIN REFERENCE | INPUT | ARCH. | SEGMENT LEARNING | PARTIAL MATCH | LOSS / TRAIN CONCEPT | RETRIEVAL DISTANCE |
|---|---|---|---|---|---|---|---|
| CQTNET | YU ET AL. (2020) | CQT | CONVNET | ✗ | ✗ | CLASSIF. | COSINE |
| DORAS&PEETERS | DORAS & PEETERS (2020) | HCQT | CONVNET | ✗ | ✗ | TRIPLET | COSINE |
| MOVE/RE-MOVE | YESILER ET AL. (2020A) | CREMA | CONVNET | ✗ | ✗ | TRIPLET | EUCLIDEAN |
| PICKINET | O'HANLON ET AL. (2021) | CQT | CONVNET | ✗ | ✗ | CLASSIF.+CENTER | COSINE |
| LYRACNET | HU ET AL. (2022) | CQT | WIDERESNET | ✗ | ✗ | CLASSIF. | COSINE |
| BYTECOVER1/2 | DU ET AL. (2022) | CQT | RESNET | ✗ | ✗ | CLASSIF.+TRIPLET | COSINE |
| COVERHUNTER | LIU ET AL. (2023) | CQT | CONFORMER | ✗ | ✓ | CLASSIF.+FOCAL+CENTER | COSINE |
| BYTECOVER3/3.5 | DU ET AL. (2023) | CQT | RESNET | ✓ | ✗ | CLASSIF.+TRIPLET | COSINE |
| DVINET/DVINET+ | ARAZ ET AL. (2024A) | CQT | CONVNET | ✗ | ✗ | TRIPLET | COSINE |
| CLEWS (PROPOSED) | THIS PAPER | CQT | RESNET | ✓ | ✓ | CONTRASTIVE | EUCLIDEAN |

For instance, up to 56% of a recent realistic large-scale data set of around $500\,\text{k}$ tracks is formed by only 2-item classes, with an average of 5 items per class (Araz et al., 2024a). This characteristic suggests that, besides the traditional focus on classification and triplet losses, a supervised contrastive learning approach (Sec. 2) could also work well.

In this paper, we consider a full music track as a succession of weakly-labeled audio segments, and learn a contrastive representation using such weak supervision. To do so, we introduce two main methods. First, we develop a number of pairwise distance selection strategies, which reduce a segment-level distance matrix into a track-level distance matrix. This enables the direct utilization of track-level annotations without statically assigning them to some or all of the segments. Second, we reformulate the alignment and uniformity (A&U) loss of Wang & Isola (2020), originally introduced for self-supervised learning, to operate on a (weakly) supervised learning task. Motivated by decoupling, hyper-parameter, and geometric considerations, we introduce several changes that convert A&U into a new loss function: the strict decoupling of positives and negatives (Yeh et al., 2022), the simplification of hyper-parameters, the native operation in Euclidean geometry (cf. Koishekenov et al., 2023), and a smoothing constant for negative pairs. With both distance reduction and contrastive learning strategies, we do not only outperform existing approaches in the segment-level evaluation by a large margin, but we also achieve state-of-the-art results in the standard track-level evaluation. We also perform an extensive ablation study to empirically compare the proposed methods with several alternatives, including additional reduction strategies and common contrastive losses. We believe that, due to the generality of the challenges addressed here, the proposed methods may find utility in further domains beyond musical version matching. To facilitate understanding and reproduction, we share our code and model checkpoints in https://github.com/sony/clews.

## 2. Background

After a history of rule-, feature-, and model-based approaches (Serrà, 2011; Yesiler et al., 2021), musical version matching is currently tackled as a supervised learning problem, focusing on full-track pairwise matching (Table 1). However, two versions do not necessarily need to match for their entire duration, and actually several applications rely on few-second partial matches. Only a couple of approaches base their learning or retrieval stages on segments or partial matches, respectively. ByteCover3 (Du et al., 2023) pioneered learning from segments with their "maxmean" operator. However, such operator still does not allow for partial matches, as it forces all segments of a track to match some segment from another track. CoverHunter (Liu et al., 2023) is able to detect partial matches of around $45\,\text{s}$. However, the learning strategy to do so is based on a two-stage brute-force approach. First, it trains a coarse detector model on 15-second segments using classification, focal, and center losses. Then, it resorts to this first-stage model and a rule-based approach to (weakly) label 45-second segments, which are finally used to train the second-stage model with the same losses. In both stages, CoverHunter treats segments as full tracks. To our knowledge, we are the first to consider an entirely segment-based approach for both learning and retrieval stages.

The literature on musical version matching has traditionally considered a number of classification (Sun et al., 2014) and triplet (Schroff et al., 2015) loss variants, and their combination (Table 1). However, given the same ground truth, another approach to learning version representations would be to consider a supervised contrastive loss like N-pairs (Sohn, 2016) or SupCon (Khosla et al., 2021). In addition, a number of well-established losses for self-supervised learning like InfoNCE/NT-Xent (Van den Oord et al., 2018; Chen et al., 2020), alignment and uniformity (Wang & Isola, 2020), or SigLIP (Zhai et al., 2023) could also be adapted. An analysis of the relations between many of such losses is

carried out by Koromilas et al. (2024). Apart from the loss function, other considerations such as positive/negative decoupling (Yeh et al., 2022) and the correspondence between distance and space geometry (Koishekenov et al., 2023) are potentially relevant in a practical case. To our knowledge, we are the first to consider a supervised contrastive loss for musical version matching.

## 3. Contrastive Learning from Weakly-Labeled Audio Segments

We now detail our approach to perform contrastive learning from weakly-labeled audio segments (CLEWS). The first part deals with track-level labels and their allocation to segment distances (we base our development on distances, but it can be easily reformulated using similarities). The second part details the contrastive loss function we use. The third part explains our architecture and training procedure.

### 3.1. Segment Distance Reduction

**Framework —** Given the $k$-th waveform segment of the $i$-th music track, $\mathbf{x}_i^k$, we compute latent representations $\mathbf{z}_i^k = \mathcal{F}(\mathbf{x}_i^k)$, where $\mathcal{F}$ represents a neural network that pools the time-varying information of the segment into a single vector (architecture details can be found in Sec. 3.3 and Appendix A). Then, for every possible pair of segments $k$ and $l$ of every possible pair of tracks $i$ and $j$, we compute their distance $\tilde{d}_{ij}^{kl}$ and obtain the distance matrix $\tilde{\mathbf{D}}$ (Fig. 1). At this point, if there are $n$ query tracks and $m$ candidate tracks with $u$ and $v$ segments[4], respectively, we have $\tilde{\mathbf{D}} \in \mathbb{R}_{\geq 0}^{nu \times mv}$. However, since labels are only provided at the track level, our binary ground truth assignments (1 for version/positive and 0 for non-version/negative) are $\mathbf{A} \in \mathbb{Z}_2^{n \times m}$. Therefore, we need some strategy to (weakly) allocate $n \times m$ labels to $nu \times mv$ segment distances.

A naïve strategy to do such allocation would be to propagate all positive/negative track assignments to all segment comparisons in the sub-rectangle $\tilde{\mathbf{D}}_{ij}$ defined by a pair of tracks $i$ and $j$. This is the approach implicitly followed by CoverHunter (Liu et al., 2023). However, besides its poor performance (Sec. 5), this strategy incurs a fundamental error, in the sense that it is teaching the model that all positive segments are 'similar' to all other positive segments, which in the case of musical versions is false (two segments could reproduce two different motives of the same song; even though it is the same song, the segments are usually not the same, unless it is an extremely repetitive song). Instead of trying to allocate $n \times m$ positive/negative distances to $nu \times mv$ segment distances, we take the opposite view

---

[4]Notice that, similar to cross-attention in Transformers, we can deal with different track lengths by taking a maximum length $u, v$ and masking. See Fig. 1 (right) for a couple of examples.

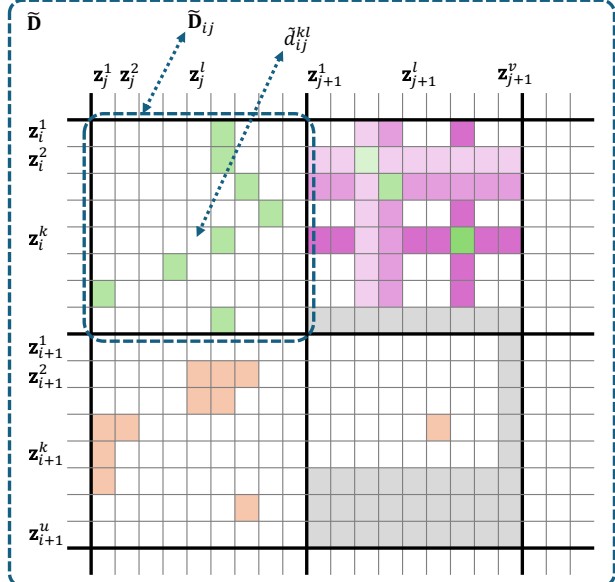

*Figure 1.* Illustration of four reduction functions $\mathcal{R}$ over pairwise segment distances $\tilde{d}_{ij}^{kl}$. They are depicted on different subrectangles $\tilde{\mathbf{D}}_{ij}$, where tracks $i$, $j$, and $j+1$ are versions (green squares) and track $i+1$ is not (orange squares). The four functions correspond to: $\mathcal{R}_{\text{meanmin}}$ (top left), $\mathcal{R}_{\text{bpwr-3}}$ (top right), $\mathcal{R}_{\text{best-10}}$ (bottom left), and $\mathcal{R}_{\text{min}}$ (bottom right). The $\mathcal{R}_{\text{bpwr-3}}$ strategy depicts its minimum/masking recursion in increasingly dark levels (green/purple cells). The sub-rectangles for $\mathcal{R}_{\text{bpwr-3}}$ and $\mathcal{R}_{\text{min}}$ also exemplify dealing with different lengths by masking (gray cells).

and reduce $nu \times mv$ segment distances to $n \times m$ track distances, matching the dimensions of positive/negative assignments $\mathbf{A}$. More specifically, we consider several reduction functions $\mathcal{R}$, producing $\mathbf{D} = \mathcal{R}(\tilde{\mathbf{D}})$, where $\mathbf{D} \in \mathbb{R}_{\geq 0}^{n \times m}$. In addition, we employ different reduction functions for positive and negative pairs, $\mathcal{R}^+$ and $\mathcal{R}^-$, respectively.

**Reduction Functions —** The naïve strategy outlined above would correspond to a mean reduction over the entire subrectangle determined by $\tilde{\mathbf{D}}_{ij}$:

$$d_{ij} = \mathcal{R}_{\text{mean}}\left(\tilde{\mathbf{D}}_{ij}\right) = \frac{1}{uv} \sum_{\substack{1 \leq k \leq u \\ 1 \leq l \leq v}} \tilde{d}_{ij}^{kl}.$$

Instead, if we just consider the $r$ best matches across segments, we have

$$d_{ij} = \mathcal{R}_{\text{best-r}}\left(\tilde{\mathbf{D}}_{ij}\right) = \frac{1}{r} \sum_{1 \leq t \leq r} \text{topr}(\tilde{\mathbf{D}}_{ij})_t$$

for $r \leq uv$, where $\text{topr}(\mathbf{D})$ is a function that returns the lowest $r$ distances in $\mathbf{D}$ (Fig. 1, bottom left). Another possibility is to consider just the single best correspondence in the entire sub-rectangle (Fig. 1, bottom right), yielding

$$d_{ij} = \mathcal{R}_{\text{min}}\left(\tilde{\mathbf{D}}_{ij}\right) = \min_{\substack{1 \leq k \leq u \\ 1 \leq l \leq v}} \tilde{d}_{ij}^{kl}.$$

A further alternative is to use an operator like the one used in ByteCover3 (Du et al., 2023), which first searches for the candidate best match per query and then averages across all query segments. Reformulating it for distances (Fig. 1, top left), we obtain

$$d_{ij} = \mathcal{R}_{\text{meanmin}}\left(\tilde{\mathbf{D}}_{ij}\right) = \frac{1}{u} \sum_{1 \leq k \leq u} \min_{1 \leq l \leq v} \tilde{d}_{ij}^{kl}.$$

Notice that the $\mathcal{R}_{\text{mean}}$, $\mathcal{R}_{\text{best-r}}$, and $\mathcal{R}_{\text{meanmin}}$ strategies above do not explicitly prevent multiple consecutive segments of track $i$ being assigned to the same segment of track $j$ (vertical or horizontal traces in Fig. 1). As mentioned, this is unrealistic for the majority of music tracks (and in general for any signal or sequence presenting a minimal variability with time). Notice also that $\mathcal{R}_{\text{mean}}$ and $\mathcal{R}_{\text{meanmin}}$ force a full-track match, that is, they are teaching the model that all segments in track $i$ should find a match in track $j$. Again, this is an unrealistic assumption for musical versions where the structure changes (and in general for any signal or sequence featuring only partial matches).

**Best-pair Reduction —** Motivated by the issues of consecutive and global segment matching above, we decide to design an additional reduction strategy that explicitly deals with both. We term it bpwr-r, for 'best pair without replacement' with a threshold $r$. In a nutshell, $\mathcal{R}_{\text{bpwr-r}}$ operates by sorting all distances in the $\tilde{\mathbf{D}}_{ij}$ sub-rectangle in increasing order, then taking the first one (say the one involving segments $k$ and $l$), removing all distances computed by using either one or the other segment (either $k$ or $l$), and iterating $r$ times. It then takes the average among those $r$ best pairwise distances. More formally, we can express it as

$$d_{ij} = \mathcal{R}_{\text{bpwr-r}}\left(\tilde{\mathbf{D}}_{ij}\right) = \frac{1}{r} \sum_{1 \leq q \leq r} \mathcal{R}_{\text{min}}\left(\tilde{\mathbf{D}}_{ij}^{(q)}\right) \quad (1)$$

for $r \leq \min(u, v)$, with the recursion

$$\tilde{\mathbf{D}}_{ij}^{(q)} = \begin{cases} \tilde{\mathbf{D}}_{ij} & \text{for } q = 1, \\ \text{maskmin}\left(\tilde{\mathbf{D}}_{ij}^{(q-1)}\right) & \text{for } q > 1, \end{cases}$$

where maskmin($\mathbf{D}$) is a function that masks the row and the column corresponding to the minimum element in $\mathbf{D}$, such that those elements are not eligible by the $\mathcal{R}_{\text{min}}$ of Eq. 1 in iterations $q > 1$. A schema of $\mathcal{R}_{\text{bpwr-3}}$ is illustrated in Fig. 1 (top right; recursion is depicted by progressively darker colors). Notice that masking rows and columns avoids the issue of consecutive segment matching[5], and that a threshold $r < \min(u, v)$ avoids the issue of full-track matching, which only happens when $r = \min(u, v)$.

---

[5]There are some specific cases where a pattern could occupy two consecutive segments (e.g., when it is split between them or when it is reproduced at half the speed). We claim that, in such cases, learning from just one of the two consecutive segments is enough (our results in Sec. 5 support this claim).

**Positives and Negatives Reduction —** We note that the previous distance reduction strategies have some conceptual parallelism with the negative/positive mining strategies used with triplet losses (Schroff et al., 2015) or in some contrastive approaches (cf. Kalantidis et al., 2020). In our case, for instance, $\mathcal{R}_{\text{min}}$ could correspond to a hard mining strategy, while $\mathcal{R}_{\text{best-r}}$ or $\mathcal{R}_{\text{bpwr-r}}$ could be regarded as semi-hard mining of segment pairs. Thus, inspired by those strategies, we decide to study if applying different reduction strategies for positives and negatives has some effect in our setup. To obtain a track-based pairwise distance matrix that combines different reductions for positive and negative pairs, we calculate

$$\mathbf{D} = \mathbf{A} \odot \mathcal{R}^{+}(\tilde{\mathbf{D}}) + (\mathbf{1} - \mathbf{A}) \odot \mathcal{R}^{-}(\tilde{\mathbf{D}}), \quad (2)$$

where $\odot$ denotes element-wise multiplication and $\mathbf{1}$ is the all-ones matrix (recall that the elements in $\mathbf{A}$ are 1 for positives and 0 otherwise). The reductions $\mathcal{R}^{+}$ and $\mathcal{R}^{-}$ can be chosen among the ones presented above.

### 3.2. Contrastive Loss

**Motivation —** After computing pairwise track-level distances $\mathbf{D}$, we need a contrastive loss that can exploit them and that, ideally, can outperform the existing losses in the considered task. For that, one can consider any supervised contrastive loss function that operates on distances, or adapt an existing self-supervised loss to the supervised framework (Sec. 2). In our case, we opt for the latter and choose the A&U loss of Wang & Isola (2020) due to its appealing properties and intuitive derivation. One of the practical properties we value is that, by using expectations, we have a similar behavior for different batch sizes (Wang & Isola 2020; see also Koromilas et al. 2024). In our analysis, we will use the concept of "potential" as introduced by Wang & Isola (2020), but nonetheless will depart from the concept of uniformity in the hypersphere.

**Changes to Alignment and Uniformity —** The A&U loss, designed for self-supervised contrastive learning, expects one positive for each item in the batch (obtained through some augmentation) while negatives correspond to all other elements in the batch. To adapt A&U to supervised contrastive learning with multiple positives per anchor, we need to carefully define both positive and negative sets. In particular, we want to preserve the decoupling of the alignment and uniformity terms as, apart from respecting the original idea of A&U, it typically yields improved performance (Yeh et al., 2022). Therefore, from all pairwise assignments $\mathbf{A}$ in the batch, we need to gather positive $A^{+}$ and negative $A^{-}$ assignment sets such that $A^{+} \cap A^{-} = \emptyset$. This also implies discarding comparisons of one track against itself (that is, the diagonals of $\mathbf{D}$ and $\mathbf{A}$, and potentially other spurious cells corresponding to sampling the same track more than once in the same batch).

Given the sets $A^+$ and $A^-$, we can write a decoupled, supervised version of A&U over a batch as

$$\tilde{\mathcal{L}} = \frac{1}{|A^+|} \sum_{(i,j) \in A^+} d_{ij}^\alpha + \lambda \log \left( \frac{1}{|A^-|} \sum_{(i,j) \in A^-} e^{-\gamma d_{ij}^2} \right),$$
(3)

where $|\ |$ denotes set cardinality and $\alpha$, $\lambda$, and $\gamma$ are hyper-parameters. Wang & Isola (2020) do not report strong performance differences by changing $\alpha$, $\lambda$, and $\gamma$ within a certain range, and generally set $\alpha = 2$, $\lambda = 1$, and $\gamma = 3$ for their experiments. In preliminary experiments, and for our task, we find similar conclusions for $\alpha$ and $\lambda$, but not for $\gamma$. In addition, we are motivated to use $\alpha = 2$ and $\lambda = 1$, as that makes alignment and uniformity terms more comparable (same distance and same weight; see also the gradient analysis below).

The original A&U loss employs Euclidean distances on the hypersphere, using $L_2$-normalized **z** vectors. This, in our view, presents a potential issue, in the sense that the employed distance function does not match with the geometric structure of the space (Euclidean vs. hypersphere surface, respectively). In our approach, instead of considering the negative arc length (which corresponds to the geodesic distance in the hypersphere) to improve performance like Koishekenov et al. (2023), we opt for the plain Euclidean space (of which the Euclidean distance is its geodesic distance). Thus, we do not constrain **z** to have a unit norm. Notice that, with this change, the uniformity concept does not apply, as we are not in an hypersphere anymore. Nonetheless, we can still reason and base our intuitions on the kernel and potential concepts used to derive the uniformity term in Wang & Isola (2020).

With the decoupling, hyper-parameter, and geometric considerations above, we formulate the CLEWS loss as

$$\mathcal{L} = \frac{1}{|A^+|} \sum_{(i,j) \in A^+} d_{ij}^2 + \log \left( \varepsilon + \frac{1}{|A^-|} \sum_{(i,j) \in A^-} e^{-\gamma d_{ij}^2} \right),$$

where $d_{ij}$ are distances after reduction (Eq. 2) and $\gamma, \varepsilon > 0$ are hyper-parameters. We use dimension-normalized Euclidean distances (root mean squared differences), as this does not affect our geometric considerations (it just adds a constant) and facilitates maintaining the same hyper-parameters when changing the dimensionality of **z**. The $\varepsilon$ hyper-parameter is initially introduced for numerical stability. However, we note that it also has a soft thresholding or smoothing effect for the potential between negative pairs.

**Role of the Hyper-parameters —** We now briefly and intuitively study the role of $\gamma$ and $\varepsilon$ in $\mathcal{L}$ (a full analysis is beyond the scope of the present paper). To do so, we consider the gradient of $\mathcal{L}$ for a specific distance pair $d_{ij}$.

Depending if the pair is in $A^+$ or $A^-$, we have

$$\nabla^+ \triangleq \frac{\partial \mathcal{L}}{\partial d_{ij}} \bigg|_{(i,j) \in A^+} = \frac{2 d_{ij}}{|A^+|}$$

or

$$\nabla^- \triangleq \frac{\partial \mathcal{L}}{\partial d_{ij}} \bigg|_{(i,j) \in A^-} = \frac{-2\gamma d_{ij} e^{-\gamma d_{ij}^2}}{|A^-|\varepsilon + c + e^{-\gamma d_{ij}^2}}, \quad (4)$$

where $e^{-\gamma d_{ij}^2}$ corresponds to the negative potential for the pair $i, j$ (Wang & Isola, 2020) and the constant $c$ is the sum of all potentials that do not feature $(i, j)$. To facilitate our analysis, we view $\varepsilon$ as a reference potential and redefine $\hat{\varepsilon} = |A^-|\varepsilon + c$. We can then consider three cases with regard to the relation of $\hat{\varepsilon}$ and the potential for the negative pair $i, j$. If $\hat{\varepsilon} \ll e^{-\gamma d_{ij}^2}$ (case 1), we have $\nabla^- \approx -2\gamma d_{ij}$ and, if $\hat{\varepsilon} \approx e^{-\gamma d_{ij}^2}$ (case 2), we have $\nabla^- \approx -\gamma d_{ij}$. In both cases, and thanks to having set $\alpha = 2$ after Eq. 3, the terms in $\nabla^-$ are similar to the ones in $\nabla^+$ (thus positive and negative pairs have a comparable influence). Moreover, we see that $\gamma$ is also acting as a weight for the negative pairs' gradient (thus taking a similar role as the original $\lambda$ in Eq. 3). Finally, if $\hat{\varepsilon} \gg e^{-\gamma d_{ij}^2}$ (case 3), we have

$$\nabla^- \approx \frac{-2\gamma d_{ij} e^{-\gamma d_{ij}^2}}{\hat{\varepsilon}},$$

which implies a progressively vanishing gradient with increasing $d_{ij}$ ($e^{-\gamma d_{ij}^2}$ decreases much faster than $\gamma d_{ij}$ increases). Thus, we obtain a smooth transition to zero gradient after the potential $e^{-\gamma d_{ij}^2}$ crosses a threshold that is a function of $\varepsilon$. A visualization of the effect of $\gamma$ and $\varepsilon$ on $\nabla^-$ (Eq. 4) is given in Appendix A.

### 3.3. Architecture and Training

We now overview CLEWS' network architecture $\mathcal{F}$ and its training procedure (further details are available in Appendix A and in our code). To obtain segment embedding vectors **z** on which to compute distances, we start from the full-track audio waveform and uniformly randomly cut a 2.5 min block **x** from it. We further cut **x** into 8 non-overlapping 20-second segments $\mathbf{x}^k$ (we repeat-pad the last segment). We then compute its constant-Q spectrogram, downsample it in time by a factor of 5, and normalize it between 0 and 1, all following similar procedures as common version matching approaches (Yesiler et al., 2021). After that, we pass it to a learnable frontend, formed by two 2D strided convolutions, batch normalization, and a ReLU activation. Next, we employ a pre-activation ResNet50 backbone (He et al., 2016) with ReZero (Bachlechner et al., 2021) and instance-batch normalization (Pan et al., 2018). We pool the remaining spectro-temporal information with generalized mean pooling (Radenović et al., 2019), and

project to a 1024-dimensional representation $\mathbf{z}$ using batch normalization and a linear layer. We train CLEWS with $\mathcal{R}^+ = \mathcal{R}_{\text{bpwr-5}}$, $\mathcal{R}^- = \mathcal{R}_{\text{min}}$, $\gamma = 5$, and $\varepsilon = 10^{-6}$ as defaults, and study the effect of such choices in Sec. 5. Since test sets also contain tracks longer than the 2.5 min used for training, in CLEWS we use our proposed $\mathcal{R}_{\text{bpwr-10}}$ for track matching, together with a segment hop size of 5 s.

We train all models with Adam using a learning rate of $2 \cdot 10^{-4}$, following a reduce-on-plateau schedule with a 10-epoch patience and an annealing factor of 0.2. The only exception is in ablation experiments, where we train for 20 epochs featuring a final 5-epoch polynomial learning rate annealing. In every epoch, we group all tracks into batches of 25 anchors and, for each of them, we uniformly sample with replacement 3 positives from the corresponding version group (excluding the anchor). Thus, we get an initial (track-based) batch size of 100. For every track in the batch, we uniformly sample 2.5 min from the full-length music track and create the aforementioned 8 segments per track. Thus, we get a final (segment-based) batch size of 800. We only use time stretch, pitch roll, and SpecAugment augmentations (Liu et al., 2023).

## 4. Evaluation Methodology

**Data —** We train and evaluate all models on the publicly-available data sets DiscogsVI-YT (DVI; Araz et al., 2024a) and SHS100k-v2 (SHS; Yu et al., 2020), using the predefined partitions. SHS is a well-established reference data set. However, since it is based on YouTube links, it is almost impossible to gather it entirely nowadays (we managed to gather 82% of it). In addition, one could consider it slightly biased, as the version group sizes are unrealistically large (cf. Doras & Peeters, 2020; Araz et al., 2024a). Instead, the recently proposed DVI data set is 5 times larger and better represents the real-world distribution of version group sizes. For both data sets, we use 16 kHz mono audio and cap the maximum length to the first 10 min.

**Baselines —** To compare the performance of the proposed approach with the state of the art, we consider several baselines: CQTNet (Yu et al., 2020), MOVE (Yesiler et al., 2020a), LyraC-Net (Hu et al., 2022), CoverHunter (Liu et al., 2023), DVINet+ (Araz et al., 2024b), Bytecover2 (Du et al., 2022), ByteCover3 (Du et al., 2023), and ByteCover3.5 (Du et al., 2024). For CoverHunter, we just consider the first "coarse" stage, as that is the part dealing with segments. CoverHunter, CQTNet, and DVINet have convenient source code available, and thus we can produce results by using it in our own pipeline (this way we can compare those baselines with our model rigorously under the same setting). Due to GPU memory restrictions, we train with randomly-sampled audio blocks of 2.5 min. For the other baselines, we can only use already reported results as reference. The ByteCover series of models are known to be non-reproducible (O'Hanlon et al., 2021; Hu et al., 2022), with all attempts to date substantially under-performing[6] the reported results. We also implement our version of the ByteCover models and, for the first time, are able to obtain results that come close to the ones reported in the original papers (Sec. 5). In the following, we denote our approximations to ByteCover with a † symbol. Having retrained/replicated baselines provides us an estimation of their performance in scenarios that have not yet been considered in the literature, such as with the DVI data set or the segment-level evaluation proposed below.

**Evaluation —** During testing, to compute candidate embeddings, we treat all models as if they were segment-based and extract overlapping blocks or segments using the same length as in training and a hop size of 5 s (this yielded a marginal improvement for full-track baselines trained on 2.5 min blocks). With these candidate embeddings, we perform both track- and segment-level evaluations. The former is equivalent to the usual evaluation setup in musical version matching, while the latter focuses on the retrieval of best-matching segments. For the track-level evaluation, we use the same segment length (the training segment length) and hop size for both queries and candidates. Then, to measure the performance of the system working at the full-track level, we use $\mathcal{R}_{\text{meanmin}}$ to compute the final query-candidate distance. For the segment-level evaluation, we keep the same segment configuration as in the track-level case, but we vary the query segment length $\tau$. This way we assess a model's performance on different query lengths found in real-world scenarios. Then, to measure the performance of the system working at the segment level, we use $\mathcal{R}_{\text{min}}$ to compute a best-match query-candidate distance. With this best-match approach, we simulate the performance of an equivalent segment-based retrieval system using all raw segments as candidates (see Appendix B). As evaluation measures, we compute the usual mean average precision (MAP) plus an enhanced version of the normalized average rank (NAR; see Appendix B). MAP focuses on the precision in the top candidates while NAR focuses more on the overall recall, which we think is a better option for musical version matching, especially for segment-based applications.

## 5. Results

**Comparison with the State of the Art —** First of all, we focus on the track-level evaluation and compare with the state of the art. We observe that CLEWS outperforms all considered approaches, many of them by a large margin (Table 2). CLEWS obtains a NAR of 2.70 on DVI-Test and a MAP of 0.876 on SHS-Test, setting a new state-of-the-art result on both data sets. Besides CLEWS, an interesting

---

[6]See for instance https://github.com/Orfium/bytecover/issues/2.

*Table 2.* Track-level evaluation and comparison with the state of the art. The symbol † denotes that it is our implementation and the ± symbol denotes 95% confidence intervals.

| APPROACH | DVI-TEST | | SHS-TEST | |
|---|---|---|---|---|
| | NAR ↓ | MAP ↑ | NAR ↓ | MAP ↑ |
| COVERHUNTER-COARSE (LIU ET AL., 2023) | $10.36 \pm 0.07$ | $0.157 \pm 0.001$ | $4.09 \pm 0.17$ | $0.491 \pm 0.007$ |
| MOVE (YESILER ET AL., 2020A) | N/A | N/A | N/A | 0.519 |
| CQTNET (YU ET AL., 2020) | $6.68 \pm 0.07$ | $0.493 \pm 0.002$ | $2.67 \pm 0.16$ | $0.677 \pm 0.007$ |
| DVINET+ (ARAZ ET AL., 2024B) | $3.69 \pm 0.06$ | $0.643 \pm 0.002$ | $2.39 \pm 0.16$ | $0.720 \pm 0.007$ |
| LYRAC-NET (HU ET AL., 2022) | N/A | N/A | N/A | 0.765 |
| BYTECOVER3† (BASED ON DU ET AL., 2023) | $5.64 \pm 0.05$ | $0.513 \pm 0.002$ | $1.91 \pm 0.14$ | $0.783 \pm 0.006$ |
| BYTECOVER1/2† (BASED ON DU ET AL., 2022) | $4.98 \pm 0.06$ | $0.595 \pm 0.002$ | $1.95 \pm 0.14$ | $0.813 \pm 0.006$ |
| BYTECOVER3 (DU ET AL., 2023) | N/A | N/A | N/A | 0.824 |
| BYTECOVER3.5 (DU ET AL., 2024) | N/A | N/A | N/A | 0.857 |
| BYTECOVER2 (DU ET AL., 2022) | N/A | N/A | N/A | 0.863 |
| CLEWS (PROPOSED) | $\mathbf{2.70 \pm 0.05}$ | $\mathbf{0.774 \pm 0.002}$ | $\mathbf{1.27 \pm 0.12}$ | $\mathbf{0.876 \pm 0.005}$ |

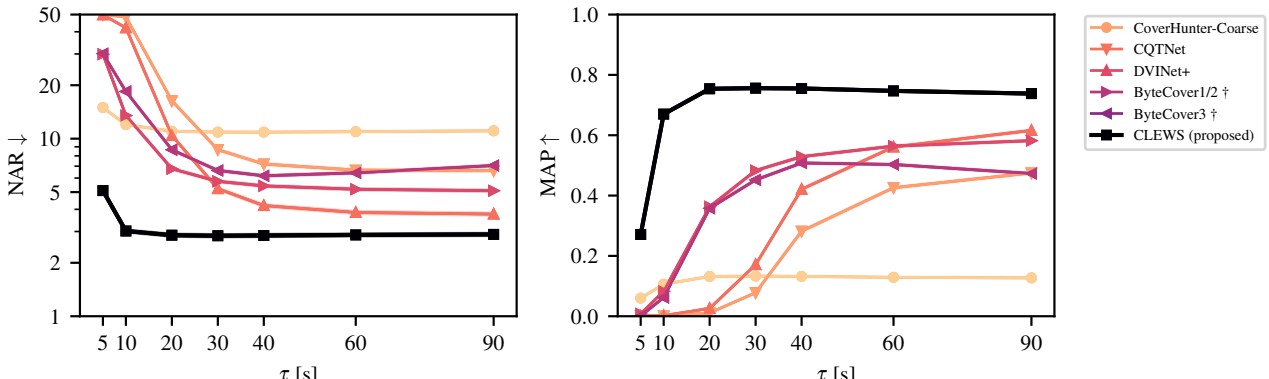

*Figure 2.* Segment-level evaluation with DVI-Test. NAR (left) and MAP (right) for different query segment lengths $\tau$ (notice the logarithmic axis for NAR). The shaded regions correspond to 95% confidence intervals (barely visible due to the size of DVI-Test). Comparatively similar results for SHS-Test and also for an alternative evaluation protocol are available in Appendix C.

observation to make is that the rank of an existing approach considerably varies between SHS-Test and DVI-Test. That is the case of DVINet+, which obtains a modest performance on SHS-Test, but achieves the best performance among the existing approaches on DVI-Test. We hypothesize that this is thanks to being the only considered approach (apart from CLEWS) that does not learn from a classification loss: because SHS-Train has more items per class than the more realistic DVI-Train (an average of 12 vs. 2, respectively), a classification loss is able to learn a useful representation from SHS-Train, but not from DVI-Train. Both DVINet+ and CLEWS, utilizing triplet and contrastive losses, respectively, do not suffer from this issue and maintain a comparatively similar performance across the two data sets. Another observation to make with regard to the consideration of segments is that treating them independently (CoverHunter-Coarse) yields worse results than developing some specific strategies (ByteCover3, ByteCover3†, and CLEWS), and that learning from a global match (ByteCover3 and ByteCover3†) is not as optimal as learning from a partial match (CLEWS). The latter is further supported by our ablations below.

**Segment-based Version Matching —** We now focus on the segment-level evaluation and study the performance as a function of the query segment length $\tau$. We observe that CLEWS again outperforms all considered models both in DVI-Test (Fig. 2) and SHS-Test (Appendix C), and for both NAR and MAP measures. Importantly, CLEWS maintains a high performance for all considered lengths (Fig. 2). The only exception is with $\tau = 5$ where, according to our listening experience, it is sometimes difficult even for a human to establish if two audio segments are versions or not. According to our segment-level evaluation, ByteCover3† features some noticeable performance degradation with large $\tau$, perhaps due to the global match approach. CQTNet and DVINet+, both based on the same plain convolutional architecture, show an early performance decline for $\tau < 60$.

**Ablations and Hyper-parameters —** Finally, we focus our attention on possible variations to the default CLEWS. We start by studying the effect of positive $\mathcal{R}^+$ and negative $\mathcal{R}^-$ segment distance reductions (Table 3). For $\mathcal{R}^+$, if we keep $\mathcal{R}^- = \mathcal{R}_{\min}$, we observe that, depending on the evaluation

*Table 3.* Results on DVI-Valid for different positive $\mathcal{R}^+$ and negative $\mathcal{R}^-$ distance reductions. The default CLEWS reductions are $\mathcal{R}^+ = \mathcal{R}_{\text{bpwr-5}}$ and $\mathcal{R}^- = \mathcal{R}_{\text{min}}$. As above, $\pm$ denotes 95% c.i.

| $\mathcal{R}^+$ | $\mathcal{R}^-$ | NAR $\downarrow$ | MAP $\uparrow$ |
|---|---|---|---|
| CLEWS (PROPOSED) | | $2.57 \pm 0.09$ | $0.804 \pm 0.003$ |
| $\mathcal{R}_{\text{bpwr-3}}$ | $\mathcal{R}_{\text{min}}$ | $2.60 \pm 0.09$ | $\mathbf{0.809 \pm 0.003}$ |
| $\mathcal{R}_{\text{bpwr-8}}$ | $\mathcal{R}_{\text{min}}$ | $\mathbf{2.51 \pm 0.09}$ | $0.789 \pm 0.003$ |
| $\mathcal{R}_{\text{meanmin}}$ | $\mathcal{R}_{\text{min}}$ | $2.58 \pm 0.09$ | $0.798 \pm 0.003$ |
| $\mathcal{R}_{\text{best-10}}$ | $\mathcal{R}_{\text{min}}$ | $2.63 \pm 0.09$ | $0.795 \pm 0.003$ |
| $\mathcal{R}_{\text{min}}$ | $\mathcal{R}_{\text{min}}$ | $2.79 \pm 0.09$ | $0.799 \pm 0.003$ |
| $\mathcal{R}_{\text{bpwr-5}}$ | $\mathcal{R}_{\text{best-10}}$ | $2.82 \pm 0.10$ | $0.779 \pm 0.003$ |
| $\mathcal{R}_{\text{bpwr-5}}$ | $\mathcal{R}_{\text{bpwr-5}}$ | $2.88 \pm 0.10$ | $0.778 \pm 0.003$ |
| $\mathcal{R}_{\text{bpwr-5}}$ | $\mathcal{R}_{\text{meanmin}}$ | $4.95 \pm 0.12$ | $0.488 \pm 0.004$ |

*Table 4.* Results on DVI-Valid for different loss functions using the default CLEWS reductions of $\mathcal{R}^+ = \mathcal{R}_{\text{bpwr-5}}$ and $\mathcal{R}^- = \mathcal{R}_{\text{min}}$.

| LOSS FUNCTION | NAR $\downarrow$ | MAP $\uparrow$ |
|---|---|---|
| CLEWS (PROPOSED) | $\mathbf{2.57 \pm 0.09}$ | $\mathbf{0.804 \pm 0.003}$ |
| SUPCON | $2.69 \pm 0.09$ | $0.676 \pm 0.004$ |
| SIGLIP | $2.79 \pm 0.09$ | $0.684 \pm 0.004$ |
| TRIPLET | $3.08 \pm 0.11$ | $0.717 \pm 0.004$ |
| SUPCON-DECOUPLED | $3.14 \pm 0.11$ | $0.739 \pm 0.004$ |
| A&U-DECOUPLED | $3.25 \pm 0.11$ | $0.620 \pm 0.004$ |
| CLASSIFICATION XENT | $8.91 \pm 0.14$ | $0.205 \pm 0.003$ |

measure, we have two options that are better than the default: $\mathcal{R}_{\text{bpwr-8}}$ for NAR and $\mathcal{R}_{\text{bpwr-3}}$ for MAP. Nonetheless, we decide to keep the default one as a compromise between the two. With such compromise in mind, we observe that, for positive reductions, global matches ($\mathcal{R}_{\text{bpwr-8}}$ and $\mathcal{R}_{\text{meanmin}}$) under-perform a partial matches ($\mathcal{R}_{\text{bpwr-r}}$), and that learning from consecutive segments ($\mathcal{R}_{\text{best-10}}$ and $\mathcal{R}_{\text{meanmin}}$) is not as competitive as avoiding them ($\mathcal{R}_{\text{bpwr-r}}$). For $\mathcal{R}^-$, if we keep $\mathcal{R}^+ = \mathcal{R}_{\text{bpwr-5}}$, we see that all considered negative reductions under-perform the default $\mathcal{R}_{\text{min}}$. We hypothesize that, as with triplet losses, a hard negative mining or worst-case strategy is beneficial (cf. Schroff et al., 2015; Kalantidis et al., 2020). Reduction strategies based on $\mathcal{R}_{\text{mean}}$ did not learn well, both for $\mathcal{R}^+$ and $\mathcal{R}^-$ (not shown).

The next aspect we study is the effect of the loss function given the default reduction strategies for $\mathcal{R}^+$ and $\mathcal{R}^-$ (Table 4). We observe that the proposed CLEWS loss performs better in both NAR and MAP, with a significant difference in the latter measure. Standard losses like SupCon, SigLIP, and Triplet (Sec. 2) come next, not being able to take as much profit from $\mathcal{R}^+$ and $\mathcal{R}^-$ as CLEWS in the task we study. As already mentioned, a standard classification loss based on cross-entropy does not perform well, especially when training with very few instances per class.

The last aspect we study is the effect of hyper-parameters $\gamma$ and $\varepsilon$. If we zoom in the resolution of NAR and MAP, we

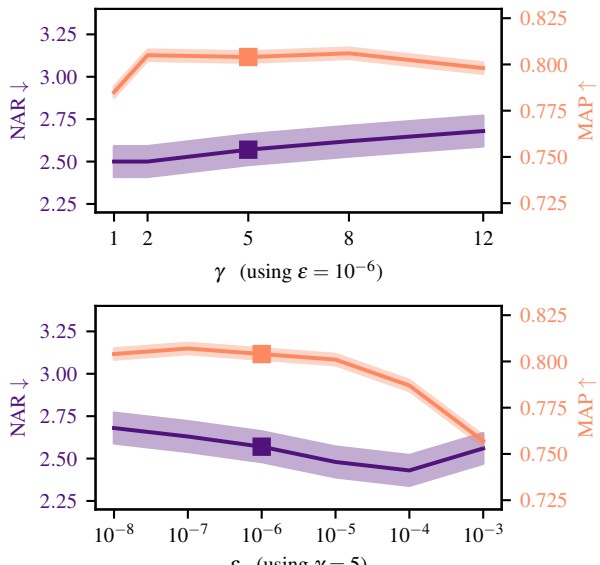

*Figure 3.* Effect of hyper-parameters $\gamma$ (top) and $\varepsilon$ (bottom) on DVI-Valid. Shaded regions correspond to 95% confidence intervals, and the default value is highlighted with a square marker.

observe an opposite trend for the two evaluation measures (Fig. 3): with progressively decreasing NAR (better performance), we obtain a progressively decreasing MAP (worse performance). This indicates that hyper-parameters $\gamma$ and $\varepsilon$ can be deliberately tuned to benefit one or the other measure (we did not extensively tune them for the results reported previously). Moreover, we actually see that setting $\gamma = 2$ could have provided a better NAR and a slighlty increased MAP. Overall, however, if we zoom ourselves out from the resolution shown in Fig. 3, we essentially observe a plateau of performance between $\gamma \in [2, 8]$ and $\varepsilon \in [10^{-8}, 10^{-5}]$.

# 6. Conclusion

In this paper, we tackle the task of segment-based musical version matching, and propose both a strategy to deal with weakly-labeled segments and a contrastive loss that outperforms well-studied alternatives. Through a series of extensive experiments, we show that our approach not only achieves state-of-the-art results in two different datasets and two different metrics, but also that it significantly outperforms existing approaches in a best-match, segment-level evaluation. We also study the effect of different reduction strategies, compare against existing losses, and analyze the effect of the hyper-parameters in our ablation studies. As weakly labeled segment information is ubiquitous in many research areas, and since the concepts exploited here are general to a wide range of contrastive learning tasks, we believe our methods could serve as inspiration or find usefulness in domains beyond audio and musical version matching.

## Acknowledgements

We thank Toshimitsu Uesaka for his comments on an earlier version of the paper. R. Oguz Araz is supported by the pre-doctoral program AGAUR-FI ajuts (2024 FI-3 00065) Joan Oró, and the Cátedras ENIA program "IA y Música: Cátedra en Inteligencia Artificial y Música" (TSI-100929-2023-1).

## Impact Statement

This paper presents work whose goal is to advance the fields of machine learning and music information retrieval. Musical version matching can be used to enhance music discovery, preserve cultural heritage, and support fair copyright management. By connecting versions across styles and performances, musical version matching also fosters creativity, promotes artistic appreciation, and paves the way for more equitable solutions in the music industry, benefiting society at large. As with any machine learning tool, however, there always exists the possibility of some potential misuses of itself or of some of its components, none of which we feel must be specifically highlighted here.

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

# APPENDIX

In this supplementary part of the paper we provide further information on the proposed method (Appendix A). We also explain with detail our evaluation methodology (Appendix B). Finally, we show additional results that could not fit in the main manuscript (Appendix C).

## A. Method Details

### A.1. Loss Gradient Visualization

In Sec. 3.2 of the main manuscript, we study the effect of hyper-parameters $\gamma$ and $\varepsilon$ on the gradient of negative pairs $\nabla^-$. Here, to further facilitate understanding, we plot the result of $\nabla^-$ (Eq. 4) for a range of potentials $e^{-\gamma d_{ij}^2}$ under different values of $\gamma$ and $\varepsilon$ in Fig. 4. We do so using $|A^-| = 128$ and $c = (|A^-| - 1)e^{-\gamma d_{ij}^2}$. With the latter, we approximate the case where the average negative potential is not far from the potential of the $i, j$ pair.

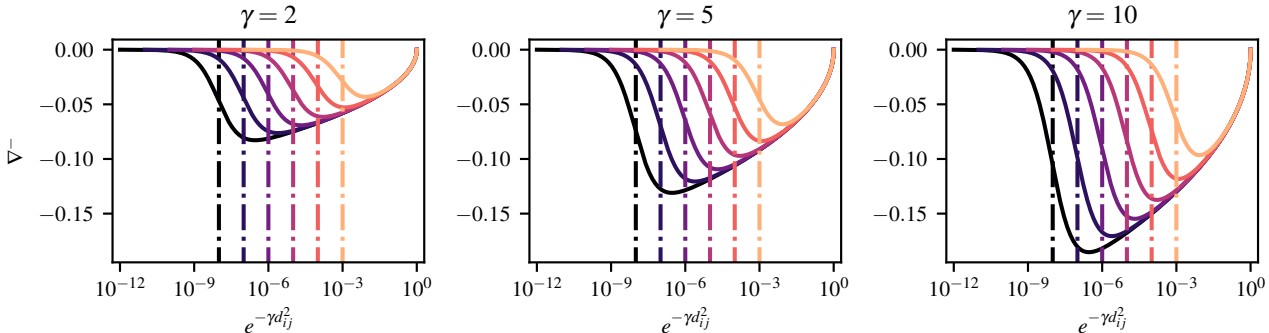

Figure 4. Plot of $\nabla^-$ as a function of the negative pair potential $e^{-\gamma d_{ij}^2}$ for different values of $\gamma$ and $\varepsilon$. From left to right, we show $\gamma = \{2, 5, 10\}$. From darker to lighter, colors correspond to $\varepsilon = \{10^{-8}, 10^{-7}, 10^{-6}, 10^{-5}, 10^{-4}, 10^{-3}\}$. Dash-dotted lines indicate each $\varepsilon$ value (notice that, in $\mathcal{L}$, $\varepsilon$ is compared to an average negative pair potential, hence placing $\varepsilon$ as a reference in the potential axis makes sense).

### A.2. A More Numerically-Friendly Version of $\mathcal{L}$

For conducting all our experiments, we found no issue in the use of $\mathcal{L}$ with regard to numerical stability with 32-bit precision. However, we should note that $\mathcal{L}$, as written in the main manuscript, may have some numerical instability, especially when employing abnormally small/large values of $\varepsilon/\gamma$, or potentially when using a numerical precision below 32 bits. In such cases, we recommend switching to the formulation below.

First of all, we multiply the terms inside the logarithm by $1/\varepsilon$:

$$\mathcal{L} = \frac{1}{|A^+|} \sum_{(i,j) \in A^+} d_{ij}^2 + \log\left(\varepsilon + \frac{1}{|A^-|} \sum_{(i,j) \in A^-} e^{-\gamma d_{ij}^2}\right) = \frac{1}{|A^+|} \sum_{(i,j) \in A^+} d_{ij}^2 + \log\left(1 + \frac{1}{\varepsilon|A^-|} \sum_{(i,j) \in A^-} e^{-\gamma d_{ij}^2}\right) + \log(\varepsilon).$$

With this, we obtain the term $\log(\varepsilon)$, which is just a constant that does not affect the gradient and can thus be dropped. Next, we perform the change of variable $1/(\varepsilon|A^-|) = \beta e^b$, where $b \geq 0$ is a constant we will set for the upper numerical limit we allow to the exponential. With this change and a few simple operations, we arrive to

$$\mathcal{L} = \frac{1}{|A^+|} \sum_{(i,j) \in A^+} d_{ij}^2 + \log\left(1 + \beta \sum_{(i,j) \in A^-} e^{b - \gamma d_{ij}^2}\right),$$

where $\beta = 1/(\varepsilon|A^-|e^b)$. We can now choose $b$ as a compromise between the overflow of $e^b$ when $d_{ij} = 0$ and the underflow of $e^{b - \gamma d_{ij}^2}$ when $d_{ij}$ is large. For normalized Euclidean distances $d$ and the ranges of $\varepsilon$ and $\gamma$ we consider, we choose $b = 10$. Note that, in addition, $\log(1 + x)$ can be implemented with `log1p(x)` in most scientific programming languages.

### A.3. Model

We here provide further specification of our network architecture (for full detail we refer the interested reader to the published code). As mentioned, we use 16 kHz mono audio with a maximum length of 10 min for both training and evaluation. For training, we cut 2.5 min blocks uniformly at random. For CLEWS, we divide such blocks into 8 non-overlapping 20-second segments. As the last segment is only 10 s, we take the opportunity to repeat-pad such segment and also consider it in our training, with the hope that this will facilitate retrieval with query lengths shorter than 20 s (which we also repeat-pad as they would not be long enough to accommodate the total striding factor of our architecture).

After obtaining segments, we apply a constant-Q transform (CQT) with 20 ms hop size, spanning 7 octaves (from a minimum frequency of 32.7 Hz), and with 12 bins per octave (we use the nnAudio library[7] in non-trainable mode, with the rest of the parameters set as default). We then take the CQT magnitude and average in time every 5 consecutive frames without overlap. This CQT representation is sent to three data augmentation functions (explained in the next subsection).

The neural network architecture starts by taking the square root of the CQT magnitude, normalizing every segment's representation between 0 and 1, and applying a learnable affine transformation. Next, we apply a 128-channel 2D convolution with a frequency-time kernel size of 12×3 and a frequency-time stride of (1,2). This is followed by batch normalization (BN), a ReLU activation, and a 256-channel 2D convolution with a kernel size of 12×3 and a stride of (2,2). This constitutes our frontend. Unless stated otherwise, we use the default PyTorch[8] parameters from version 2.3.1.

As mentioned in the main manuscript, our backbone is formed by pre-activation ResNet modules with ReZero and instance-batch normalization (IBN). We use 3, 4, 6, and 3 residual blocks with 256, 512, 1024, and 2048 channels, respectively. The strides are (1,1), (2,2), (2,2), and (1,1) for each block. The residual blocks have an IBN–ReLU–conv–BN–ReLU–conv structure, with a kernel of 3×3 in the convolution layers. To reduce GPU memory consumption, we employ half the channel dimension inside the residual block. If there is some channel or stride change, the skip connection features a BN-ReLU-conv block also with a 3×3 kernel.

The output of the backbone is time- and frequency-pooled by a generalized mean pooling operation with a single learnable exponent. Finally, the result is processed with BN and projected to 1024 dimensions by a linear layer. None of our linear or convolutional layers feature bias terms. As mentioned in the main manuscript, we use normalized squared Euclidean distances (mean squared differences) between embedding vectors.

### A.4. Training

We train all models with Adam using the default PyTorch parameters, a learning rate of $2 \cdot 10^{-4}$, and a batch size of 800 segments chosen from 100 tracks featuring 3 positives per anchor. In the main experiments, we follow a reduce-on-plateau strategy for the learning rate, monitoring an average between MAP and NAR measures on the validation set. We define an epoch as using all training tracks as anchor once, and set a 10-epoch patience period and an annealing factor of 0.2. Using this strategy, training CLEWS on SHS and DVI takes approximately 2 and 9 days, respectively, using two NVIDIA H100-80GB GPUs. In the ablation experiments, to reduce the computational burden, we only train for 20 epochs and, during the last 5 epochs, we apply a polynomial learning rate annealing with an exponent of 2.

During training, we employ three CQT data augmentation functions: SpecAugment, time stretch, and pitch roll. For SpecAugment, we mask a maximum of 15% of the time/frequency tiles. For time stretch, we resample by a uniformly sampled factor between 0.6 and 1.8. For pitch roll, we choose a uniform value between $-12$ and $+12$. In the DVI data set, we use a probability of 0.1 independently for each augmentation. However, since the SHS data set is considerably smaller than DVI and potentially features less variability, we find some benefit in increasing such probability for SHS. In that case, we set the augmentation probabilities to 0.4, 0.3, and 0.5 for SpecAugment, time stretch, and pitch roll, respectively.

## B. Evaluation Methodology Details

### B.1. Track- and Segment-level Evaluations

For the track-level evaluation, we cut the entire raw waveform (up to the first 10 min) into overlapping blocks or segments using a hop size of 5 s. For both the queries and the candidates, the length of such blocks/segments corresponds to the same

---

[7] https://github.com/KinWaiCheuk/nnAudio
[8] https://pytorch.org/docs/2.3/

length we used to train each model (that is, 2.5 min for CQTNet, DVINet+, and ByteCover1/2† and 20 s for CoverHunter, ByteCover3†, and CLEWS). Next, we compute pairwise distances for each query-candidate block/segment and apply a distance reduction function. We use $\mathcal{R}_{\text{meanmin}}$ for all models except CLEWS, which exploits the newly proposed $\mathcal{R}_{\text{bpwr-10}}$. After reduction we obtain a track-based distance matrix that we can use to sort candidates per query and compute common evaluation measures.

For the segment-level evaluation, we also cut the entire raw waveform into overlapping blocks/segments with a hop size of 5 s. For candidates, we also use the same length that we used to train each model (same as in the track-level evaluation). However, for the queries, we extract multiple-length segments with a hop size of 5 s (we consider segment lengths $\tau = \{5, 10, 20, 30, 40, 60, 90\}$ s). Then, given a segment length, we compute pairwise distances for each query-candidate block/segment, and apply the $\mathcal{R}_{\text{min}}$ distance reduction to obtain a track-based distance matrix. After that, the evaluation proceeds as with the track-level evaluation (and any common evaluation protocol in musical version matching). The usage of $\mathcal{R}_{\text{min}}$ puts the focus on the best-matching segment per track, and is equivalent to performing version matching on an index formed by all possible segments, treating them independently, and removing duplicate track names after sorting.

### B.2. Normalized Average Rank

To evaluate retrieval performance and complement mean average precision (MAP), we employ an enhanced version of the normalized average rank (NAR), originally proposed by Müller et al. (2001). Given a list of retrieved items $R$, sorted in descending order of predicted relevance to a query $q$, and containing a set of target matches $M = \{m_1, \ldots m_{|M|}\}$, $M \subset R$, Bosteels & Kerre (2007) redefined NAR as

$$\widetilde{\text{NAR}}_q = \frac{1}{|M||R|} \sum_{i=1}^{|M|} \Big( \text{rank}(m_i, R) - i \Big),$$

where the function $\text{rank}(m, R) \in [1, |R|]$ returns the rank of $m$ in $R$. This definition, as well as the one of Müller et al. (2001), yields 0 for perfect retrieval, 0.5 for random retrieval, and approaches 1 as performance worsens. However, a value equal to one is never obtained. Not only that, but the maximum bound inversely depends on $|M|$ and, therefore, can be different for each query $q$. To avoid that, one should replace the number of retrieved items $|R|$ in the denominator by the number of non-relevant retrieved items $|R| - |M|$. Hence, we correct the definition of Bosteels & Kerre (2007) and employ

$$\text{NAR}_q = \frac{100}{|M| \, (|R| - |M|)} \sum_{i=1}^{|M|} \Big( \text{rank}(m_i, R) - i \Big),$$

which additionally yields a convenient % value, now between 0 and 100 for all sizes of $M$. Our final number is the average over all queries $Q$:

$$\text{NAR} = \frac{1}{|Q|} \sum_{q \in Q} \text{NAR}_q.$$

Note that, in research evaluation scenarios, one must compute both MAP and NAR measures excluding the query from the candidate list.

## C. Additional Results

### C.1. Segment-level Evaluation with the Best Match Protocol

In the main manuscript, we present the results for the segment-level evaluation on DVI-Test (Fig. 2). The exact numbers for such plots can be found here in Table 5. For SHS-Test, we obtain comparable results, which can be found below in Fig. 5 and Table 6.

### C.2. Segment-level Evaluation with the Random Segment Protocol

In our segment-level evaluation, we adopt a best match protocol as specified in Sec. 4. However, Du et al. (2023) introduced what could be termed as the 'random segment' protocol: "For each query, we constructed a query set consisting of the original full-track recording, and 9 music clips randomly cut from it, with the duration being 6, 10, 15, 20, 25, 30, 40, 50 and 60 seconds respectively" (Du et al., 2023). Apart from lacking further specification, we claim that using random segments

biases the evaluation, as we can never reach a perfect accuracy (a random segment from a song does not necessarily need to have a match in a version song). Furthermore, if the objective is to match tracks by their segments, we believe using random segments for evaluation may implicitly favor approaches exploiting more generic or global track characteristics than the specific matching-segment information. These are the reasons why we introduce our segment-based protocol. However, in the spirit of comparing with existing reported values, and to avoid any doubt on the performance of the proposed approach, we replicate such protocol (to our best) and compute again results for all methods considered here. They are shown in Fig. 6 and Table 7 below, together with the MAP values of ByteCover2, ByteCover3, and Re-MOVE (Yesiler et al., 2020b) reported by Du et al. (2023).

## C.3. Runtime

To conclude, we also provide an informal runtime analysis for the considered models (Table 8). For a fair comparison, inference times are measured with the segment-based evaluation protocol, thus all models perform the same task of segment-based retrieval with a 5 s hop size, using $\mathcal{R}_{\min}$. We should also note that the time complexity of the naïve implementation of the reductions studied above is $O(uv)$ for $\mathcal{R}_{\text{mean}}$ and $\mathcal{R}_{\min}$, $O(uv + u)$ for $\mathcal{R}_{\text{meanmin}}$, $O(r + uv \log(uv))$ for $\mathcal{R}_{\text{best-r}}$, and $O(r(uv + u + v))$ for $\mathcal{R}_{\text{bpwr-r}}$, where $r \leq \min(u, v)$ and $u, v$ are the number of considered segments in a sub-rectangle (see main text). Note that the values for $r$, $u$, and $v$ are small for today's computation standards. For instance, a 5 min song with 20 s segments and no overlap yields $u = 15$.

*Table 5.* Segment-level evaluation with DVI-Test. NAR (top) and MAP (bottom) results for different lengths of query segments $\tau$. The $\pm$ symbol marks 95% confidence intervals.

| APPROACH | $\tau$ [s] | | | | | | |
|---|---|---|---|---|---|---|---|
| | 5 | 10 | 20 | 30 | 40 | 60 | 90 |
| COVERHUNTER-COARSE | $14.97 \pm 0.07$ | $12.01 \pm 0.07$ | $11.01 \pm 0.07$ | $10.89 \pm 0.07$ | $10.88 \pm 0.07$ | $10.95 \pm 0.07$ | $11.07 \pm 0.07$ |
| CQTNET | $49.96 \pm 0.10$ | $48.48 \pm 0.10$ | $16.35 \pm 0.08$ | $8.67 \pm 0.07$ | $7.20 \pm 0.07$ | $6.65 \pm 0.07$ | $6.60 \pm 0.07$ |
| DVINET+ | $49.80 \pm 0.13$ | $42.11 \pm 0.12$ | $10.42 \pm 0.07$ | $5.23 \pm 0.06$ | $4.20 \pm 0.06$ | $3.84 \pm 0.06$ | $3.76 \pm 0.06$ |
| BYTECOVER1/2 † | $29.91 \pm 0.10$ | $13.50 \pm 0.08$ | $6.77 \pm 0.06$ | $5.75 \pm 0.06$ | $5.42 \pm 0.06$ | $5.19 \pm 0.06$ | $5.09 \pm 0.06$ |
| BYTECOVER3 † | $30.11 \pm 0.10$ | $18.45 \pm 0.08$ | $8.66 \pm 0.06$ | $6.62 \pm 0.06$ | $6.18 \pm 0.06$ | $6.42 \pm 0.06$ | $7.06 \pm 0.06$ |
| CLEWS (OURS) | $5.10 \pm 0.06$ | $3.02 \pm 0.05$ | $2.86 \pm 0.05$ | $2.84 \pm 0.05$ | $2.85 \pm 0.05$ | $2.87 \pm 0.05$ | $2.89 \pm 0.05$ |
| COVERHUNTER-COARSE | $0.060 \pm 0.001$ | $0.106 \pm 0.001$ | $0.132 \pm 0.001$ | $0.133 \pm 0.001$ | $0.132 \pm 0.001$ | $0.129 \pm 0.001$ | $0.127 \pm 0.001$ |
| CQTNET | $0.001 \pm 0.000$ | $0.001 \pm 0.000$ | $0.011 \pm 0.000$ | $0.078 \pm 0.001$ | $0.282 \pm 0.002$ | $0.426 \pm 0.002$ | $0.475 \pm 0.002$ |
| DVINET+ | $0.001 \pm 0.000$ | $0.002 \pm 0.000$ | $0.026 \pm 0.000$ | $0.171 \pm 0.001$ | $0.421 \pm 0.002$ | $0.561 \pm 0.002$ | $0.616 \pm 0.002$ |
| BYTECOVER1/2 † | $0.008 \pm 0.000$ | $0.083 \pm 0.001$ | $0.363 \pm 0.002$ | $0.483 \pm 0.002$ | $0.529 \pm 0.002$ | $0.564 \pm 0.002$ | $0.582 \pm 0.002$ |
| BYTECOVER3 † | $0.001 \pm 0.000$ | $0.062 \pm 0.001$ | $0.358 \pm 0.002$ | $0.452 \pm 0.002$ | $0.508 \pm 0.002$ | $0.503 \pm 0.002$ | $0.473 \pm 0.002$ |
| CLEWS (OURS) | $0.271 \pm 0.002$ | $0.670 \pm 0.002$ | $0.754 \pm 0.002$ | $0.756 \pm 0.002$ | $0.755 \pm 0.002$ | $0.747 \pm 0.002$ | $0.738 \pm 0.002$ |

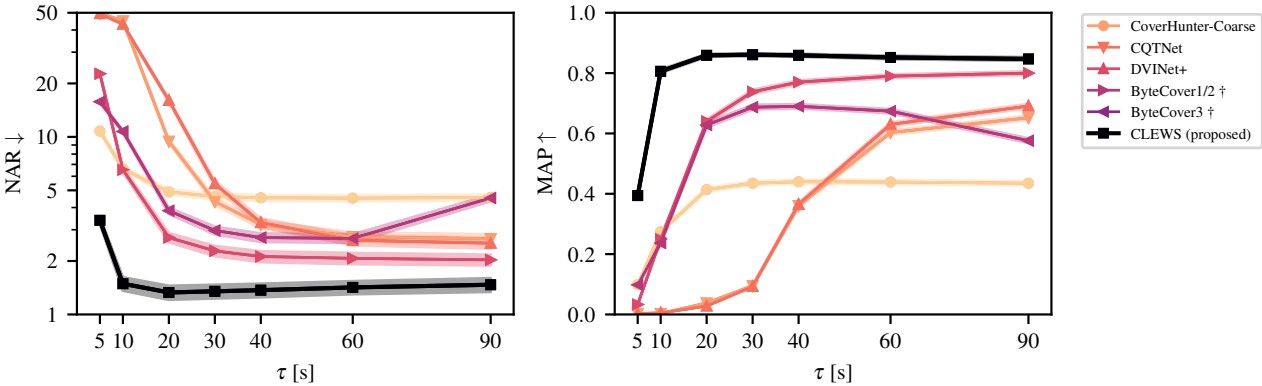

*Figure 5.* Segment-level evaluation with SHS-Test. NAR (left) and MAP (right) for different lengths of query segments $\tau$ (notice the logarithmic axis for NAR). The shaded regions correspond to 95% confidence intervals.

*Table 6.* Segment-level evaluation with SHS-Test. NAR (top) and MAP (bottom) results for different lengths of query segments $\tau$. The $\pm$ symbol marks 95% confidence intervals.

| Approach | $\tau$ [s] | | | | | | |
|---|---|---|---|---|---|---|---|
| | 5 | 10 | 20 | 30 | 40 | 60 | 90 |
| CoverHunter-Coarse | $10.78 \pm 0.23$ | $6.61 \pm 0.20$ | $4.90 \pm 0.18$ | $4.62 \pm 0.18$ | $4.54 \pm 0.18$ | $4.52 \pm 0.18$ | $4.56 \pm 0.18$ |
| CQTNet | $49.80 \pm 0.48$ | $44.98 \pm 0.47$ | $9.45 \pm 0.25$ | $4.32 \pm 0.18$ | $3.21 \pm 0.16$ | $2.74 \pm 0.16$ | $2.67 \pm 0.16$ |
| DVINet+ | $49.44 \pm 0.52$ | $43.12 \pm 0.52$ | $16.05 \pm 0.36$ | $5.45 \pm 0.22$ | $3.29 \pm 0.18$ | $2.62 \pm 0.17$ | $2.52 \pm 0.17$ |
| ByteCover1/2 † | $22.68 \pm 0.42$ | $6.53 \pm 0.22$ | $2.71 \pm 0.16$ | $2.28 \pm 0.15$ | $2.12 \pm 0.15$ | $2.07 \pm 0.15$ | $2.03 \pm 0.14$ |
| ByteCover3 † | $15.78 \pm 0.32$ | $10.73 \pm 0.20$ | $3.84 \pm 0.16$ | $2.97 \pm 0.15$ | $2.71 \pm 0.15$ | $2.68 \pm 0.15$ | $4.53 \pm 0.19$ |
| CLEWS (ours) | $3.39 \pm 0.17$ | $1.49 \pm 0.13$ | $1.33 \pm 0.12$ | $1.35 \pm 0.12$ | $1.37 \pm 0.12$ | $1.42 \pm 0.12$ | $1.47 \pm 0.13$ |
| CoverHunter-Coarse | $0.099 \pm 0.004$ | $0.274 \pm 0.007$ | $0.414 \pm 0.007$ | $0.435 \pm 0.007$ | $0.440 \pm 0.007$ | $0.439 \pm 0.007$ | $0.435 \pm 0.007$ |
| CQTNet | $0.003 \pm 0.000$ | $0.003 \pm 0.000$ | $0.038 \pm 0.001$ | $0.095 \pm 0.003$ | $0.361 \pm 0.007$ | $0.603 \pm 0.007$ | $0.652 \pm 0.007$ |
| DVINet+ | $0.003 \pm 0.000$ | $0.005 \pm 0.000$ | $0.028 \pm 0.001$ | $0.093 \pm 0.003$ | $0.365 \pm 0.007$ | $0.630 \pm 0.007$ | $0.691 \pm 0.007$ |
| ByteCover1/2 † | $0.033 \pm 0.002$ | $0.250 \pm 0.006$ | $0.640 \pm 0.007$ | $0.738 \pm 0.007$ | $0.770 \pm 0.006$ | $0.790 \pm 0.006$ | $0.800 \pm 0.006$ |
| ByteCover3 † | $0.098 \pm 0.004$ | $0.237 \pm 0.007$ | $0.628 \pm 0.008$ | $0.687 \pm 0.008$ | $0.690 \pm 0.007$ | $0.674 \pm 0.007$ | $0.576 \pm 0.008$ |
| CLEWS (ours) | $0.394 \pm 0.007$ | $0.806 \pm 0.006$ | $0.859 \pm 0.005$ | $0.861 \pm 0.005$ | $0.859 \pm 0.005$ | $0.852 \pm 0.006$ | $0.847 \pm 0.006$ |

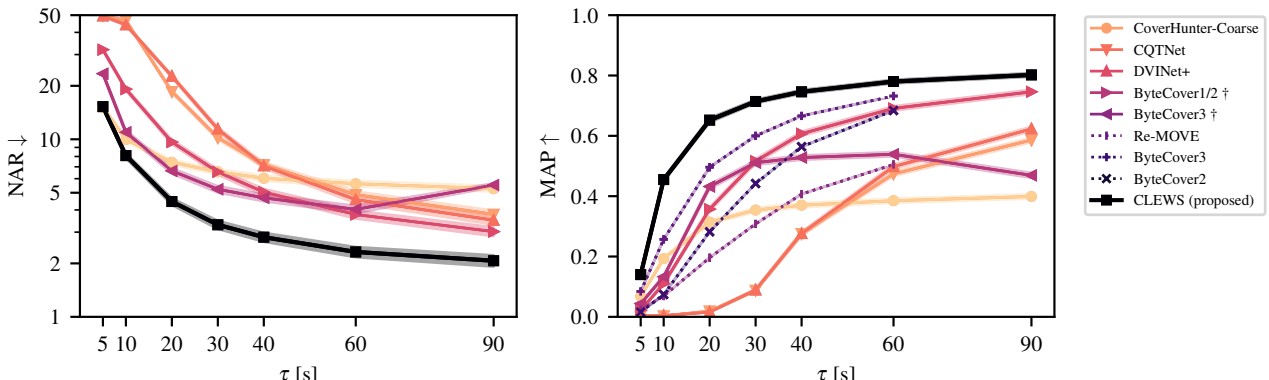

*Figure 6.* Segment-level evaluation with SHS-Test using the random segment protocol of Du et al. (2023). NAR (left) and MAP (right) for different lengths of random query segments $\tau$ (notice the logarithmic axis for NAR). The shaded regions correspond to 95% confidence intervals. The dotted lines correspond to values reported by Du et al. (2023).

*Table 7.* Segment-level evaluation with SHS-Test using the random segment protocol of Du et al. (2023). NAR (top) and MAP (bottom) results for different lengths of random query segments $\tau$. The $\pm$ symbol marks 95% confidence intervals. Du et al. (2023) did not report any confidence interval.

| Approach | $\tau$ [s] | | | | | | |
|---|---|---|---|---|---|---|---|
| | 5 | 10 | 20 | 30 | 40 | 60 | 90 |
| CoverHunter-Coarse | $15.15 \pm 0.30$ | $9.97 \pm 0.25$ | $7.42 \pm 0.22$ | $6.56 \pm 0.21$ | $6.03 \pm 0.21$ | $5.62 \pm 0.20$ | $5.27 \pm 0.19$ |
| CQTNet | $49.73 \pm 0.48$ | $46.67 \pm 0.47$ | $18.55 \pm 0.32$ | $10.21 \pm 0.25$ | $7.18 \pm 0.23$ | $4.88 \pm 0.20$ | $3.78 \pm 0.18$ |
| DVINet+ | $49.50 \pm 0.52$ | $44.13 \pm 0.51$ | $22.65 \pm 0.38$ | $11.41 \pm 0.28$ | $7.09 \pm 0.24$ | $4.57 \pm 0.20$ | $3.50 \pm 0.19$ |
| ByteCover1/2 † | $31.96 \pm 0.40$ | $19.16 \pm 0.34$ | $9.65 \pm 0.26$ | $6.55 \pm 0.22$ | $5.04 \pm 0.21$ | $3.77 \pm 0.19$ | $3.02 \pm 0.17$ |
| ByteCover3 † | $23.41 \pm 0.37$ | $10.95 \pm 0.28$ | $6.64 \pm 0.23$ | $5.23 \pm 0.21$ | $4.68 \pm 0.20$ | $4.03 \pm 0.19$ | $5.52 \pm 0.12$ |
| CLEWS (ours) | $15.27 \pm 0.30$ | $8.09 \pm 0.25$ | $4.46 \pm 0.19$ | $3.30 \pm 0.17$ | $2.81 \pm 0.16$ | $2.32 \pm 0.15$ | $2.07 \pm 0.15$ |
| CoverHunter-Coarse | $0.068 \pm 0.003$ | $0.193 \pm 0.005$ | $0.314 \pm 0.006$ | $0.354 \pm 0.007$ | $0.370 \pm 0.007$ | $0.385 \pm 0.007$ | $0.399 \pm 0.007$ |
| CQTNet | $0.003 \pm 0.000$ | $0.003 \pm 0.000$ | $0.019 \pm 0.001$ | $0.088 \pm 0.003$ | $0.276 \pm 0.006$ | $0.474 \pm 0.007$ | $0.586 \pm 0.007$ |
| DVINet+ | $0.003 \pm 0.000$ | $0.004 \pm 0.000$ | $0.016 \pm 0.001$ | $0.089 \pm 0.003$ | $0.276 \pm 0.006$ | $0.498 \pm 0.007$ | $0.623 \pm 0.007$ |
| ByteCover1/2 † | $0.022 \pm 0.001$ | $0.110 \pm 0.004$ | $0.357 \pm 0.006$ | $0.516 \pm 0.007$ | $0.607 \pm 0.007$ | $0.691 \pm 0.007$ | $0.746 \pm 0.007$ |
| ByteCover3 † | $0.044 \pm 0.002$ | $0.133 \pm 0.004$ | $0.432 \pm 0.007$ | $0.511 \pm 0.007$ | $0.528 \pm 0.007$ | $0.539 \pm 0.007$ | $0.469 \pm 0.008$ |
| Re-MOVE | $0.023$ | $0.069$ | $0.196$ | $0.308$ | $0.407$ | $0.505$ | N/A |
| ByteCover3 | $0.084$ | $0.257$ | $0.496$ | $0.600$ | $0.666$ | $0.732$ | N/A |
| ByteCover2 | $0.016$ | $0.074$ | $0.282$ | $0.442$ | $0.564$ | $0.684$ | N/A |
| CLEWS (ours) | $0.140 \pm 0.004$ | $0.455 \pm 0.006$ | $0.652 \pm 0.007$ | $0.714 \pm 0.006$ | $0.746 \pm 0.006$ | $0.780 \pm 0.006$ | $0.802 \pm 0.006$ |

*Table 8.* Training and inference runtimes (informal) using a single NVIDIA H100 GPU. Training runtime is measured using a batch construction as specified in the main text (2.5 min audio blocks, 25 anchors, 3 positives per anchor, total of 100 audio blocks). Inference runtime is measured following the segment-based evaluation protocol (5 s hop size, $\tau = 20$ s, $\mathcal{R} = \mathcal{R}_{\min}$). Retrieval time corresponds to a database with 2000 candidates, and grows linearly with them for all approaches.

| APPROACH | PARAMETERS | TRAINING [S/BATCH] | INFERENCE | |
|---|---|---|---|---|
| | | | EMBEDDING [MS/SONG] | RETRIEVAL [MS/QUERY] |
| COVERHUNTER-COARSE | 28 M | 0.68 | 22.5 | 2.9 |
| CQTNET | 35 M | 0.48 | 57.6 | 2.0 |
| DVINET+ | 11 M | 0.38 | 60.9 | 2.2 |
| BYTECOVER3 † | 969 M | 0.71 | 27.4 | 5.1 |
| BYTECOVER1/2 † | 202 M | 0.50 | 62.2 | 5.1 |
| CLEWS | 199 M | 1.19 | 40.3 | 3.5 |

