# OpenReview forum: "Supervised Contrastive Learning from Weakly-Labeled Audio Segments for Musical Version Matching"
_ICML.cc/2025/Conference — ICML 2025 poster_

### Official Review · Reviewer_phRw · 2025-03-12

**Overall Recommendation:** 4

**Summary:**

The paper addresses the challenge of matching different renditions of the same musical piece at segment level. The authors propose a method that uses pairwise segment distance reductions for weakly annotated data, combined with a modified alignment-and-uniformity contrastive loss. This approach achieves state-of-the-art track-level results and a breakthrough in segment-level evaluation

**Claims And Evidence:**

Yes. The claims are well supported by the experiment results.

**Essential References Not Discussed:**

None.

**Experimental Designs Or Analyses:**

Yes. The experiments include both track-level and segment-level evaluations and compare with baselines. The paper also includes ablation studies on the impact of various reduction functions, loss functions, and hyperparameters.

**Methods And Evaluation Criteria:**

Yes. The authors propose methods (best-pair distance reductions + a modified contrastive loss CLEWS) that directly address the need to match musical versions at the segment level and track level. Their evaluation metrics MAP and NAR are make sense for the task.

**Other Comments Or Suggestions:**

None

**Other Strengths And Weaknesses:**

Strengths:
1. Handling short segments and showing improved results at both track and segment levels.
2. Proposes a novel supervised contrastive loss for detecting musical versions

Weaknesses:
1. It would be beneficial to include a detailed analysis comparing running times across different methods.
2. The experimental evaluation could be further strengthened by adding results on two additional widely used datasets: Covers80 and Da-tacos.

**Questions For Authors:**

1. The experimental evaluation could be further strengthened by adding results on two additional widely used datasets: Covers80 and Da-tacos.
2. (Optinal) It would be beneficial to include a detailed analysis comparing running times across different methods.

**Relation To Broader Scientific Literature:**

The paper’s contributions are mainly in the following two parts:
1. Segment-level music match approaches: although some recent studies have considered fine-grained analyses beyond track level, like ByteCover3 (Du et al., 2023) and CoverHunter (Liu et al., 2023), fully segment-based approaches to musical version detection are lacked. This paper is the first to consider an entirely segment-based approach for both learning and retrieval stages.
2. Novel supervised contrastive loss: earlier methods (CQTNET, DORAS&PEETERS, MOVE/RE-MOVE, PICKINET, LYRACNET, BYTECOVER, COVERHUNTER, DVINET/DVINET+) often used classification-style losses or triplet losses like shown in Table 1; instead, the authors introduce a supervised variant of alignment and uniformity specifically for this task.

**Theoretical Claims:**

Yes, I reviewed the CLEWS loss, especially the decoupling contrastive loss into alignment and uniformity terms and the gradient analysis involving γ and ε. I did not encounter any obvious mistakes.

---

> ### Author Rebuttal · Authors · 2025-03-28
>
> We are grateful to the Reviewer for his/her assessment and comments. We now answer and discuss about the main questions raised in the Review.
>
> 1) Additional data sets ---- This is a relevant suggestion that is worth discussing. We originally computed the results for covers80 and we paste them below. However, we finally decided to not include them in the manuscript due to the multiple issues and the low representativity of covers80. Covers80 is a very small data set, consisting of only 80 tracks (as a reference, DVI-Test contains around 100k tracks). The small size of covers80, together with the fact that different existing approaches train with different existing datasets in order to report results on covers80, makes the data set problematic and non-reliable as a reference for performance measurement (this can also be seen with the high confidence intervals reported for covers80). The Da-TACOS data set comes with pre-extracted features, which difficults a fair comparison with audio-input methods due to different feature extraction pipelines and data augmentation procedures. Like covers80, Da-TACOS does not provide a train set, which additionally creates issues for fair comparison as models are typically not trained with the same data. Finally, and related to the latter, our internal analysis shows that around 90% of Da-TACOS is contained in the DVI data set. Thus, we consider that using DVI is preferable (and enough) for comparing approaches.
>
> 2) Analysis comparing runtimes ---- Thanks to the Reviewers' suggestion we performed both a complexity and a runtime analysis, and we will include them in the Appendix of the manuscript. For convenience, we reproduce the runtime results in the answer to Q1 from Reviewer ySSi, who asked a similar question, and we refer to it for further details due to space limitation. In summary, we do not see any big issue that would severely limit the utilization of the proposed approach in front of the other segment-based evaluated approaches.
>
> ---
> Table: Track-level evaluation on the Covers80 test set. We distinguish between models trained on DVI and models trained on SHS. The +- symbol denotes 95% confidence intervals.
> ```
> Approach                        Trained on DVI               Trained on SHS
>                                 NAR         MAP              NAR         MAP
> - - - - - - - - - - - - - - - - - - - - - - - - - - - - - - - - - - - - - - - - - - - -
> CoverHunter-Coarse (our impl)   7.85+-2.37  0.558+-0.068     5.80+-2.52  0.735+-0.061
> CQTNet (reported)               n/a         n/a              n/a         0.840
> CQTNet (our impl)               3.33+-2.05  0.869+-0.048     2.10+-1.14  0.861+-0.050
> DVINet+ (our impl)              0.88+-0.51  0.897+-0.043     1.95+-1.35  0.889+-0.045
> ByteCover3 (our impl)           1.13+-0.88  0.900+-0.043     0.96+-0.75  0.916+-0.038
> ByteCover3 (reported)           n/a         n/a              n/a         0.927
> ByteCover2 (reported)           n/a         n/a              n/a         0.928
> ByteCover1/2 (our impl)         1.46+-0.93  0.905+-0.042     1.67+-1.18  0.929+-0.038
> CLEWS (proposed)                0.72+-0.63  0.955+-0.030     1.31+-1.11  0.944+-0.034
> ```

---

> > ### Comment · Reviewer_phRw · 2025-04-04
> >
> > Thanks for your rebuttal. The additional results have addressed my concern. I will keep the 'accept' recommendation.

---

### Official Review · Reviewer_ysSi · 2025-03-13

**Overall Recommendation:** 4

**Summary:**

In this paper, authors propose a method (CLEWS) to learn from weakly annotated segments, together with a contrastive loss variant that outperforms well studied alternatives. State-of-the-art performance on both track-level and segment-level evaluations, significantly outperforming existing methods. The approach addresses the mismatch between track-level annotations and segment-level inference needs in real-world applications.

**Claims And Evidence:**

1) The effect of hyperparameters on the loss function is presented in fig3 and appendix a, which alleviates my concerns regarding the potential excessive sensitivity of the loss function to hyperparameter variations.

2) The superiority of CLEWS over existing methods (table 2, appendixc, fig. 2) is supported by extensive experiments on two datasets (DVI and SHS).

3) The effectiveness of bpwr-r reduction and modified A&U loss is validated through ablation studies in table3 and table4

**Essential References Not Discussed:**

I have thoroughly reviewed the theoretical foundations presented in the paper, along with their associated references, and found no instances of related work that were overlooked or inadequately discussed. The ideas and contributions claimed in the paper are original and do not appear to be derived from prior works without proper attribution. Furthermore, the experiments conducted in the study are compared against several state-of-the-art (SOTA) methods from recent literature, ensuring a fair and up-to-date evaluation of the proposed approach.

**Experimental Designs Or Analyses:**

I have carefully reviewed the experimental setup proposed in the paper and found it to be well-designed and comprehensive. The experiments include detailed analyses of hyperparameters, and the descriptions of data augmentation techniques and evaluation protocols are both sufficient and specific. Additionally, it would be beneficial to include a brief statement regarding the computational resources consumed during training (e.g., GPU memory usage, such as 24GB) to provide further clarity on the practical requirements for reproducing the results.

**Methods And Evaluation Criteria:**

Strgenths:
1) Segment reduction strategies (e.g., bpwr-r) effectively address partial matching and avoid consecutive segment biases.
2) The modified loss decouples positives/negates and simplifies hyperparameters, aligning with contrastive learning principles.

Evaluation:
1) MAP and NAR are appropriate metrics for retrieval tasks. The corrected NAR formula (Appendix B) improves interpretability.
2) Segment-level evaluation protocols (best-match vs. random-segment) provide comprehensive insights.

**Other Comments Or Suggestions:**

Add a fig to describe the CLEWS model structure and the pipeline

**Other Strengths And Weaknesses:**

Strengths:
Practical impact: Addresses real-world needs for segment-level matching in copyright detection and generative model attribution.

Weaknesses:
Limited discussion on computational costs (e.g., bpwr-r’s runtime for large-scale datasets).

**Questions For Authors:**

Q1: How does the computational cost of bpwr-r scale with track length (e.g., 10-min vs. 2.5-min tracks)? Would this limit real-world deployment?
Impact: A high runtime for bpwr-r would weaken practicality claims.

Q2: Have you tested CLEWS on different kinds musical to assess generalizability not only focus on the testset.
Impact: Positive results would strengthen the claim of broader applicability.

Q3：In the context of music version matching, certain musical pieces are expressed in diverse forms, such as the same lyrics being rendered in distinct musical styles or rappers performing familiar songs with different beats. Can the proposed approach, which leverages distance metrics and contrastive learning, effectively distinguish such fine-grained variations in music versions? This raises questions about the method's ability to capture subtle differences in musical expression, particularly when the underlying content (e.g., lyrics) remains consistent while the musical form (e.g., rhythm, instrumentation) varies significantly.

Q4：Do the DVI and SHS datasets used in the current experiments sufficiently support training models for verifying song plagiarism and evaluating AI-generated music? While I acknowledge the challenges in collecting a more objective and larger dataset, this limitation should still be recognized as a constraint of the current work.

**Relation To Broader Scientific Literature:**

The work builds on contrastive learning (Wang & Isola, 2020), triplet losses (Schroff et al., 2015), and version matching (Serra`, 2011; Yesiler et al., 2021). Key innovations include adapting contrastive learning (A&U loss ) to supervised settings and introducing segment reduction for weak supervision, advancing partial matching in music retrieval.

**Theoretical Claims:**

I have check the theoretical proofs presented in the paper and found them to be logically sound and free from inconsistencies.

---

> ### Author Rebuttal · Authors · 2025-03-28
>
> We thank a lot the Reviewer for his/her assessment and constructive feedback. We answer to and discuss on the questions posed by the Reviewer below. As suggested by the Reviewer, we are considering adding a figure to describe the CLEWS model structure and pipeline, especially along with the shared code to facilitate understanding.
>
> 1) Computational cost ---- This is a good suggestion. Given track lengths of $N$ and $M$, the computational cost of $R_{\text{bpwr-r}}$ is $O(NM+N+M)$. Essentially, like the cost of the other reductions considered in the paper, it is quadratic (considering $N\approx M$). We will add the cost of all reductions to the Appendix. It should be noted that, with non-overlapping 20-second segments (the configuration we use for training), $N$ and $M$ are relatively small. For instance, for a 5-minute song we get $N=15$ segments, which is orders of magnitude smaller than the typical length we usually see in other quadratic and sequential operations like the ones performed, for instance, in Transformer's cross-/self-attention. Regarding computational resources used during training, we used two NVIDIA H100 80GB GPUs. We will add this information to the manuscript. Regarding retrieval runtime, we want to clarify that all approaches, when operating at a segment level, incur a natural increase as compared to track-based retrieval. Thanks to the Reviewers' suggestion, we collected these numbers and will report them in the Appendix. We paste them below for convenience. In summary, we do not see any big issue that would severely limit the utilization of the proposed approach in front of the other evaluated segment-based alternatives.
>
> 2) Different kinds of musical material --- In contrast to the SHS data set, which essentially features pop and rock songs, the DVI data set contains a wide variety of musical genres and material (from jazz standards to electronic versions, including musicals, rock, reggae, etc.). The test set of DVI is very large compared to existing evaluation standards, including over 100k different tracks. We believe that DVI-Test is representative enough to assess the generalizability of the approach with regard to western contemporary music.
>
> 3) Fine-grained variations in music versions --- We believe the DVI data set is representative enough of the real world, including cases such as the ones outlined by the Reviewer (for instance the one in which lyrics remain consistent while the musical form varies significantly). Our approach, like the vast majority of version matching approaches, focuses on the audio content, and it is true that it may underperform in cases where only the lyrics, but not the melody, are preserved. For those cases, which are not the most common ones for musical versions (according to what we see in the available data sets and sources), including a lyrics transcription system and combining with CLEWS could be a fruitful approach.
>
> 4) Song plagiarism --- This is an interesting aspect. Song plagiarism is a controversial and multifaceted topic in music copyright infringement. Often times, court decisions on music plagiarism cases are not only rooted on musical concepts and similarity, but go beyond that by considering external factors such as the possibility of access to the source by the offender, the amount of money involved, the resources employed or the collaborating artists, etc. Focusing on similarity alone, we believe the current data sets, and especially DVI, have the potential to start supporting comprehensive similarity searches, at least for western contemporary music (with the exception perhaps of some niche genres like experimental music).
>
> ---
> Table: Training and inference runtimes (informal) using a single NVIDIA H100 GPU. Training runtime is measured using a batch construction as specified in the main text (2.5 min audio blocks, 25 anchors, 3 positives per anchor, total of 100 audio blocks). Inference runtime is measured following the segment-based evaluation protocol (5 s hop size, $\tau=20$ s, $R=R_\text{min}$). Retrieval time corresponds to a database with 2000 candidates, and grows linearly with candidates.
> ```
> Approach             Parameters   Training [s/batch]                  Inference
>                                                        Embedding [ms/song]   Retrieval [ms/query]
> - - - - - - - - - - - - - - - - - - - - - - - - - - - - - - - - - - - - - - - - - - - - - - - - -
> CoverHunter-Coarse   28M          0.68                 22.5                  2.9
> CQTNet               35M          0.48                 57.6                  2.0
> DVINet+              11M          0.38                 60.9                  2.2
> ByteCover3           969M         0.71                 27.4                  5.1
> ByteCover1/2         202M         0.50                 62.2                  5.1
> CLEWS                199M         1.19                 40.3                  3.5
> ```

---

### Official Review · Reviewer_s9r9 · 2025-03-14

**Overall Recommendation:** 4

**Summary:**

This paper presents a novel approach for musical version matching at the segment level using weakly labeled audio segments and a supervised contrastive learning framework. The authors highlight two key contributions: (1) a method to effectively learn from weakly annotated musical segments, which enables more granular version matching beyond track-level approaches, and (2) a modified contrastive loss function that improves retrieval performance by incorporating decoupling, hyper-parameter tuning, and geometric considerations. The proposed method achieves state-of-the-art performance at the track level and delivers a significant breakthrough in segment-level evaluations. The generalizability of this approach suggests potential applications beyond music, such as other weakly labeled retrieval tasks.

**Claims And Evidence:**

The claims made in the paper are well-supported by empirical evidence, particularly in demonstrating improvements in both track-level and segment-level musical version matching. The proposed contrastive loss function is compared against existing triplet and classification-based losses, and the reported results indicate superior performance. However, further clarity on the sensitivity of hyper-parameters in different datasets would strengthen the claim of generalizability. Additionally, it would be beneficial if the authors provided an ablation study to isolate the contributions of each component separately.

**Essential References Not Discussed:**

None

**Experimental Designs Or Analyses:**

The experimental design is solid, with clear comparisons to existing methods. The results convincingly demonstrate the superiority of the proposed approach, but additional experiments on unseen datasets or different genres of music would further support the generalizability claim. It would also be beneficial to include confidence intervals or statistical significance tests to reinforce the reliability of the reported improvements.

**Methods And Evaluation Criteria:**

The methods and evaluation criteria appear sound and well-aligned with the problem at hand. The choice of contrastive learning is appropriate given the retrieval-based nature of the task. The benchmark datasets used for evaluation are suitable, although further discussion on the robustness of weakly labeled annotations would be helpful. It would also be valuable to see a qualitative analysis of retrieval cases where the proposed method succeeds or fails, to provide deeper insights into its behavior.

**Other Comments Or Suggestions:**

None

**Other Strengths And Weaknesses:**

Strengths:

1. The paper addresses a well-motivated problem with clear practical applications.
2. The proposed approach outperforms prior state-of-the-art methods in both track-level and segment-level evaluation.
3. The use of weakly labeled data is an important contribution that has broader applicability beyond musical version matching.
4. The contrastive loss modifications are well-motivated and lead to clear performance gains.

Weaknesses:

1. The impact of weakly labeled data quality on model performance is not deeply analyzed.
2. The paper could benefit from more qualitative examples to illustrate cases where the method succeeds or fails.
3. The generalizability claim would be stronger with additional experiments on varied datasets beyond music retrieval.

**Questions For Authors:**

1. Potential Issues with Weakly Labeled Data Assumptions. The method assumes that weakly labeled data is sufficiently informative for training, but in many cases, weak labels can introduce ambiguities (e.g., segments may contain overlapping themes from different songs).
If a weak label incorrectly associates a segment with a version it does not belong to, how does the model handle such inconsistencies?
2. Would the proposed contrastive loss modifications be beneficial for other retrieval tasks, such as speech or video?
3. Have you considered integrating self-supervised learning techniques in addition to weak supervision?
4. Can you provide further analysis on failure cases and potential biases in retrieval?
5. Ablation Studies on Loss Modifications Are Limited. The proposed contrastive loss function is modified in multiple ways, but the individual contributions of each modification are not clearly separated. An ablation study isolating each modification (decoupling, geometric tuning, etc.) would clarify which aspects contribute most to performance.
6. Segment-Level Evaluation Requires More Justification. While the paper claims a “breakthrough” in segment-level version matching, there is little discussion on the limitations of segment-based evaluation. Segment retrieval can be highly sensitive to boundary effects—how does the model perform when segments are slightly misaligned?
7.

**Relation To Broader Scientific Literature:**

The paper builds upon prior work in contrastive learning, weakly supervised learning, and music retrieval. It appropriately references fundamental works in these areas but could benefit from additional citations related to self-supervised learning in music information retrieval.

**Theoretical Claims:**

None

---

> ### Author Rebuttal · Authors · 2025-03-28
>
> We thank the Reviewer for his/her assessment of our work and the constructive feedback. We first answer to the questions posed by the Reviewer and later discuss on a few further aspects that appear in the main review.
>
> 1) Label inconsistencies ---- Like the big majority of machine learning models, our model assumes a low (or nonexistent) number of noisy labels. Hence, it does not explicitly handle song label inconsistencies. In the weak label assignment, most of our reduction strategies try to assign the song label to the best (lowest distance) segments in a pairwise comparison, while discarding the rest of the comparisons. According to the obtained results, we believe that the model ends up assigning the weak labels to the correct segments pairs (otherwise we would not be obtaining good performance). Nonetheless, it is true that perhaps one could implement further mechanisms to explicitly handle possible inconsistencies. We plan to actively consider this situation in the next version of our approach.
>
> 2) Contrastive loss modifications ---- We think our modifications to the contrastive loss could be beneficial for other retrieval tasks, and we think the examples provided by the Reviewer are perhaps the ones that have more potential, as they also correspond to domains where sequence information is important. As mentioned to Reviewer oJzQ, the extension of the proposed approach to other domains is out of the scope of the current manuscript, but nonetheless a very interesting direction for future work.
>
> 3) Self-supervised learning ---- We did consider the use of self-supervised learning techniques in addition to weak supervision. However, we did not have a clear idea on which self-supervised/proxy task to include. Predicting future segments as an aid to weak labeling was one task we played with during our experimentation, but we were not able to obtain any meaningful performance increase. We also studied additional data augmentations to create different views of the segments (we share them in the code), but again found no benefit when evaluating with the largest data set we consider (DVI). We hypothesize that such data set is large enough to not benefit from trivial data augmentations, and that the variability it naturally includes is more informative than the artificial augmentations we introduced.
>
> 4) Analysis of failure cases ---- This is a good suggestion, and we are studying which is the best way to include such analysis in the manuscript or the Appendix. Unfortunately, we have not developed a demo of the system yet. Informally, we can comment that it is hard to find a clear denominator for failure cases, as most of the time we do not see a clear or systematic failure that we could address in a further development. For instance, some true positives show a spectacular robustness towards genre variations, while still we find similar genre variations of the original song among the false negatives.
>
> 5) Ablation of loss modifications ---- We agree with the Reviewer. It was our original intention to include ablations for every loss modification. However, we finally did not do it due to two main reasons. On the one hand, the decoupling modification introduces just a marginal improvement while, in the other hand, we cannot isolate the other modifications as each one is conceptually and practically related to the other. For instance, the introduction of the smooth threshold $\varepsilon$ is needed because the sum of exponential terms inside the logarithm can become zero, and the latter is due to switching from cosine to Euclidean distance, which can grow indefinitely and cause a numerical error. Therefore, we cannot study the effect of switching from cosine to Euclidean without introducing the soft smoothing threshold $\varepsilon$. We can however study the effect of different values of the latter, which we do in Sec. 5.
>
> 6) Segment-level boundary effects ---- We think we do not fully understand the question, but we will nonetheless try to address it the better we can. In the way we perform the segment-based evaluation, segments are extracted before concatenating all candidate songs, and segment embeddings (computed independently) are then stacked together in the candidate database. At the retrieval stage, we know which is the song identifier for each segment embedding, and can thus avoid mixing different songs. Since segment embeddings are computed independently, we also avoid boundary effects with other segments and songs.
>
> Regarding confidence intervals, we want to mention that we do report 95% confidence intervals in all our results (both tables and figures). We will add further clarification in the Table captions to avoid such confusion. Regarding additional citations to self-supervised literature in music information retrieval, we are totally open to include them. If the Reviewer has specific examples in mind, we will consider them in the next revision of the manuscript.

---

> > ### Comment · Reviewer_s9r9 · 2025-04-08
> >
> > I would like to thank the authors for their comments, and based on the answers would like the keep the same score.

---

### Official Review · Reviewer_oJzQ · 2025-03-20

**Overall Recommendation:** 3

**Summary:**

The proposed paper is about cover song detection or musical version matching task. This task is to identify different versions of a same song, for example, if there exist an original song and this song has been sung by other singers with different vocal styles, this can be regarded as a cover song. Also, if there exist some electronic or sped-up remix of the original song, that's also considered as a different version. So, the task is to detect these different versions of a song in a single category. To do that, most of the previous researches conducted song or track-level matching methods. So, they compared the full two audio and measure the similarity. However, in this work, the authors tried to compare the segments of the two audio and proposed aggregation method over segment-to-segment similarity matrix. They compared few possible methods and among that the proposed aggregation method, best-pair without replacement, reached the SOTA level performance on this task. To training segment-level similarity measure network, they used contrastive learning with weak labels. So, if the two audio are in the same version, then they regarded all the segments in the songs are similar even though there exist some possibility that the two segments have different characteristics (e.g. different section with different styles) even through they are in the same version. Overall, the paper is well-written and easy-to-read. However, I personally think more contribution might be needed to have solid novelty.

I think the core novelty of the proposed method lies in segment-level similarity learning with the proposed aggregation method. However, as the authors noted that, this can be regarded as one of a triplet sampling method, or it can also be viewed as one of the multiple instance learning method. Therefore, if the main contribution lies in the method, then more experimental results might be needed on datasets with different modalities, etc.

## update after rebuttal
As the authors replied that their method is novel on soft sampling method within a song, I changed the decision to weak accept. Thanks for the hard work.

**Claims And Evidence:**

Yes, the experimental results supports the superiority of the proposed method.

**Essential References Not Discussed:**

It's good.

**Experimental Designs Or Analyses:**

More experiments with datasets with different modalities might enhance the versatility of the proposed method. Also, if more analysis on some examples, like how the actual two audio are compared or highlighted in the proposed aggregation method might be helpful for the readers.

**Methods And Evaluation Criteria:**

The authors used the conventional evaluation methods for the cover song identification task, so it's good.

**Other Comments Or Suggestions:**

-

**Other Strengths And Weaknesses:**

Dive into the segment-level similarity comparison for cover song detection itself is one that need to be conquered in the future, so motivation is good.

**Questions For Authors:**

-

**Relation To Broader Scientific Literature:**

Multiple-instance learning using segment-level similarity learning with instance-level aggregation method might be the one for broader readers.

**Theoretical Claims:**

Multiple-instance learning using segment-level similarity learning with instance-level aggregation method seems solid.

---

> ### Author Rebuttal · Authors · 2025-03-28
>
> We thank the Reviewer for his/her comments and insight. We want to clarify two misunderstandings by the Reviewer, and comment on the further analyses part. We think that the Reviewer may increase his/her score once these two misunderstandings are clarified, as they are the only two strong criticisms we could find in the review.
>
> 1) The Reviewer mentions that the method regards all the segments in the songs as similar ("then they regarded all the segments in the songs are similar") ---- We want to stress that it is actually the opposite. As mentioned in the manuscript, considering all compared segments between two songs as similar corresponds to the $R_{\text{mean}}$ strategy. We clearly motivate why this is unrealistic for music (Sec. 3.1) and further corroborate that models do not learn well with this reduction (Sec. 5). In addition, we have to note that all reduction strategies developed after $R_{\text{mean}}$ go precisely in the direction of avoiding the situation where "all segments in the songs are similar". In particular, the proposed $R_{\text{bpwr-r}}$ reduction explicitly avoids that by construction.
>
> 2) The Reviewer mentions that the proposed approach lacks novelty, "as the authors noted that, this can be regarded as one of a triplet sampling method, or it can also be viewed as one of the multiple instance learning method." ---- We believe that this is not true, and we want to state that we do not explicitly mention that. On the one hand, none of the proposed techniques corresponds to triplet sampling, nor we use any triplet loss (except for showing better results with the proposed loss). On the other hand, the fact that we try to conceptually relate some of the reduction techniques we propose (e.g., $R_{\text{min}}$) to concepts employed in other multiple-instance learning methods (e.g., hard mining of examples) does not mean that we use a similar technique or implementation. Actually, we propose $R_{\text{bpwr-r}}$, which we do not find a clear parallelism with hard, semi-hard, or random instance mining. Besides, the development of the proposed loss is novel and does not employ triplet sampling or existing multiple instance learning methods.
>
> Regarding further analyses with different modalities, we want to recall that our main focus is on the audio retrieval field, and that the manuscript is submitted to the "Applications" track, using the "Application-Driven Machine Learning" tag. We believe a number of the techniques we employ could be transferred to other domains as they do not include audio-specific processing. However, demonstrating it in a rigorous way would require much more experimentation and results which are beyond the scope of the current manuscript. Regarding the analysis of some specific examples and on how two audio recordings are compared, we are studying to possibility to add an additional figure to the manuscript, as this was also mentioned by another reviewer. We thank the Reviewer for this suggestion.

---

### Decision · Program_Chairs · 2025-05-01

**Decision:**

Accept (poster)

**Comment:**

The paper presents a novel method to train a model for musical version matching using weakly-labeled audio segments.

- Strengths
	- Exploiting weakly-annotated segments for the training
	- Proposes novel supervised contrastive loss for music version matching
	- The proposed method shows superior performance over previous methods.

Besides the strengths, the reviewers suggested further qualitative case studies, more rigorous ablation studies, or applying it to other tasks than music version matching, although these suggestions would be difficult to be included in a single paper.

Overall, this is a strong paper with clear application and technical contribution.